# LEARNING TO UNDERSTAND GOAL SPECIFICATIONS BY MODELLING REWARD

**Dzmitry Bahdanau**[*]
Mila, Université de Montréal
dimabgv@gmail.com

**Felix Hill**
DeepMind

**Jan Leike**
DeepMind

**Edward Hughes**
DeepMind

**Arian Hosseini**
Mila, Université de Montréal

**Pushmeet Kohli**
DeepMind

**Edward Grefenstette**[†]
DeepMind
egrefen@fb.com

## ABSTRACT

Recent work has shown that deep reinforcement-learning agents can learn to follow language-like instructions from infrequent environment rewards. However, this places on environment designers the onus of designing language-conditional reward functions which may not be easily or tractably implemented as the complexity of the environment and the language scales. To overcome this limitation, we present a framework within which instruction-conditional RL agents are trained using rewards obtained not from the environment, but from reward models which are jointly trained from expert examples. As reward models improve, they learn to accurately reward agents for completing tasks for environment configurations—and for instructions—not present amongst the expert data. This framework effectively separates the representation of what instructions require from how they can be executed. In a simple grid world, it enables an agent to learn a range of commands requiring interaction with blocks and understanding of spatial relations and under-specified abstract arrangements. We further show the method allows our agent to adapt to changes in the environment without requiring new expert examples.

## 1 INTRODUCTION

Developing agents that can learn to follow user instructions pertaining to an environment is a longstanding goal of AI research (Winograd, 1972). Recent work has shown deep reinforcement learning (RL) to be a promising paradigm for learning to follow language-like instructions in both 2D and 3D worlds (e.g. Hermann et al. (2017); Chaplot et al. (2018), see Section 4 for a review). In each of these cases, being able to reward an agent for successfully completing a task specified by an instruction requires the implementation of a full interpreter of the instruction language. This interpreter must be able to evaluate the instruction against environment states to determine when reward must be granted to the agent, and in doing so requires full knowledge (on the part of the designer) of the semantics of the instruction language relative to the environment. Consider, for example, 4 arrangements of blocks presented in Figure 1. Each of them can be interpreted as

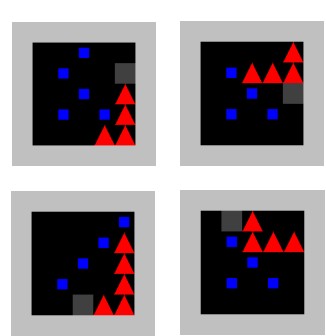

Figure 1: Different valid goal states for the instruction "build an L-like shape from red blocks".

a result of successfully executing the instruction "build an L-like shape from red blocks", despite the fact that these arrangements differ in the location and the orientation of the target shape, as well as in the positioning of the irrelevant blue blocks. At best (e.g. for instructions such as the

---

[*]Work done during an internship at DeepMind.
[†]Now at Facebook AI Research.

aforementioned one), implementing such an interpreter is feasible, although typically onerous in terms of engineering efforts to ensure reward can be given—for any admissible instruction in the language—in potentially complex or large environments. At worst, if we wish to scale to the full complexity of natural language, with all its ambiguity and underspecification, this requires solving fundamental problems of natural language understanding.

If instruction-conditional reward functions cannot conveniently or tractably be implemented, can we somehow learn them in order to then train instruction-conditional policies? When there is a single implicit task, Inverse Reinforcement Learning (IRL; Ng & Russell, 2000; Ziebart et al., 2008) methods in general, and Generative Adversarial Imitation Learning (Ho & Ermon, 2016) in particular, have yielded some success in jointly learning reward functions from expert data and training policies from learned reward models. In this paper, we wish to investigate whether such mechanisms can be adapted to the more general case of jointly learning to understand language which specifies task objectives (e.g. instructions, goal specifications, directives), and use such understanding to reward language-conditional policies which are trained to complete such tasks. For simplicity, we explore a facet of this general problem in this paper by focussing on the case of declarative commands that specify sets of possible goal-states (e.g. "arrange the red blocks in a circle."), and where expert examples need only be goal states rather than full trajectories or demonstrations, leaving such extensions for further work. We introduce a framework—Adversarial Goal-Induced Learning from Examples (AGILE)—for jointly training an instruction-conditional reward model using expert examples of completed instructions alongside a policy which will learn to complete instructions by maximising the thus-modelled reward. In this respect, AGILE relies on familiar RL objectives, with free choice of model architecture or training mechanisms, the only difference being that the reward comes from a learned reward model rather than from the environment.

We first verify that our method works in settings where a comparison between AGILE-trained policies with policies trained from environment reward is possible, to which end we implement instruction-conditional reward functions. In this setting, we show that the learning speed and performance of A3C agents trained with AGILE reward models is superior to A3C agents trained against environment reward, and comparable to that of true-reward A3C agents supplemented by auxiliary unsupervised reward prediction objectives. To simulate an instruction-learning setting in which implementing a reward function would be problematic, we construct a dataset of instructions and goal-states for the task of building colored orientation-invariant arrangements of blocks. On this task, without us ever having to implement the reward function, the agent trained within AGILE learns to construct arrangements as instructed. Finally, we study how well AGILE's reward model generalises beyond the examples on which it was trained. Our experiments show it can be reused to allow the policy to adapt to changes in the environment.

## 2 ADVERSARIAL GOAL-INDUCED LEARNING FROM EXAMPLES

Here, we introduce AGILE ("Adversarial Goal-Induced Learning from Examples", in homage to the adversarial learning mechanisms that inspire it), a framework for jointly learning to model reward for instructions, and learn a policy from such a reward model. Specifically, we learn an instruction-conditional *policy* $\pi_\theta$ with parameters $\theta$, from a data stream $\mathcal{G}^{\pi_\theta}$ obtained from interaction with the environment, by adjusting $\theta$ to maximise the expected total reward $R_\pi(\theta)$ based on stepwise reward $\hat{r}_t$ given to the policy, exactly as done in any normal Reinforcement Learning setup. The difference lies in the source of the reward: we introduce an additional discriminator network $D_\phi$, the *reward model*, whose purpose is to define a meaningful reward function for training $\pi_\theta$. We jointly learn this reward model alongside the policy by training it to predict whether a given state $s$ is a goal state for a given instruction $c$ or not. Rather than obtain positive and negative examples of ⟨instruction, state⟩ pairs from a purely static dataset, we sample them from a policy-dependent data stream. This stream is defined as follows: positive examples are drawn from a fixed dataset $\mathcal{D}$ of instructions $c_i$ paired with goal states $s_i$; negative examples are drawn from a constantly-changing buffer of states obtained from the policy acting on the environment, paired with the instruction given to the policy. Formally, the policy is trained to maximize a return $R_\pi(\theta)$ and the reward model is trained to minimize a

cross-entropy loss $L_D(\phi)$, the equations for which are:

$$R_\pi(\theta) = \mathop{\mathbb{E}}_{(c,s_{1:\infty})\sim\mathcal{G}^{\pi_\theta}} \sum_{t=1}^{\infty} \gamma^{t-1}\hat{r}_t + \alpha H(\pi_\theta), \tag{1}$$

$$L_D(\phi) = \mathop{\mathbb{E}}_{(c,s)\sim\mathcal{B}} -\log(1 - D_\phi(c,s)) + \mathop{\mathbb{E}}_{(c_i,g_i)\sim\mathcal{D}} -\log D_\phi(c_i,g_i). \tag{2}$$

where

$$\hat{r}_t = [D_\phi(c,s_t) > 0.5]$$

In the equations above, the Iverson Bracket $[\dots]$ maps truth to 1 and falsehood to 0, e.g. $[x > 0] = 1$ iff $x > 0$ and 0 otherwise. $\gamma$ is the discount factor. With $(c, s_{1:\infty}) \sim \mathcal{G}^{\pi_\theta}$, we denote a state trajectory that was obtained by sampling $(c, s_0) \sim \mathcal{G}$ and running $\pi_\theta$ conditioned on $c$ starting from $s_0$. $\mathcal{B}$ denotes a replay buffer to which $(c, s)$ pairs from $T$-step episodes are added; i.e. it is the undiscounted occupancy measure over the first $T$ steps. $D_\phi(c, s)$ is the probability of $(c, s)$ having a positive label according to the reward model, and thus $[D_\phi(c, s_t) > 0.5]$ indicates that a given state $s_t$ is more likely to be a goal state for instruction $c$ than not, according to $D$. $H(\pi_\theta)$ is the policy's entropy, and $\alpha$ is a hyperparameter. The approach is illustrated in Fig 2. Pseudocode is available in Appendix A. We note that Equation 1 differs from a traditional RL objective only in that the modelled reward $\hat{r}_t$ is used instead of the ground-truth reward $r_t$. Indeed, in Section 3, we will compare policies trained with AGILE to policies trained with traditional RL, simply by varying the reward source from the reward model to the environment.

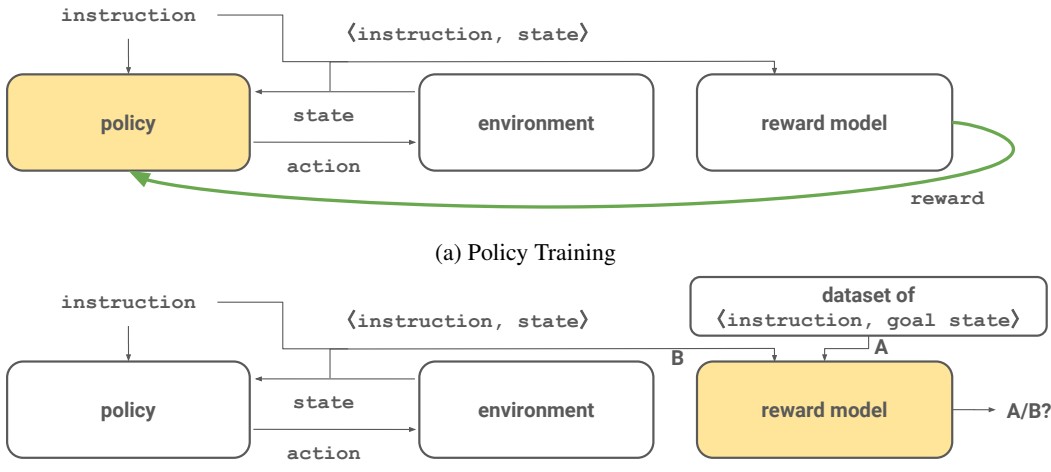

(a) Policy Training

(b) Reward Model Training

Figure 2: Information flow during AGILE training. The policy acts conditioned on the instruction and is trained using the reward from the reward model (Figure 2a). The reward model is trained, as a discriminator, to distinguish between "A", the ⟨instruction, goal-state⟩ pairs from the dataset (Figure 2b), and "B", the ⟨instruction, state⟩ pairs from the agent's experience.

**Dealing with False Negatives** Let us call $\Gamma(c)$ the objective set of goal states which satisfy instruction $c$ (which is typically unknown to us). Compared to the ideal case where all $(c, s)$ would be deemed positive if-and-only-if $s \in \Gamma(c)$, the labelling of examples implied by Equation 2 has a fundamental limitation when the policy performs well. As the policy improves, by definition, a increasing share of $(c, s) \in \mathcal{B}$ are objective goal-states from $\Gamma(c)$. However, as they are treated as negative examples in Equation 2, the discriminator accuracy drops, causing the policy to get worse. We therefore propose the following simple heuristic to rectify this fundamental limitation by approximately identifying the false negatives. We rank $(c, s)$ examples in $\mathcal{B}$ according to the reward model's output $D_\phi(c, s)$ and discard the top $1 - \rho$ percent as potential false negatives. Only the other $\rho$ percent are used as negative examples of the reward model. Formally speaking, the first term in Equation 2 becomes $\mathbb{E}_{(c,s)\sim\mathcal{B}_{D_\phi,\rho}} -\log(1 - D_\phi(c, s))$, where $\mathcal{B}_{D_\phi,\rho}$ stands for the $\rho$ percent of $\mathcal{B}$

selected, using $D_\phi$, as described above. We will henceforth refer to $\rho$ as the anticipated negative rate. Setting $\rho$ to 100% means using $\mathcal{B}_{D_\phi,100} = \mathcal{B}$ like in Equation 2, but our preliminary experiments have shown clearly that this inhibits the reward model's capability to correctly learn a reward function. Using too small a value for $\rho$ on the other hand may deprive the reward model of the most informative negative examples. We thus recommend to tune $\rho$ as a hyperparameter on a task-specific basis.

**Reusability of the Reward Model**  An appealing advantage of AGILE is the fact that the reward model $D_\phi$ and the policy $\pi_\theta$ learn two related but distinct aspects of an instruction: the reward model focuses on recognizing the goal-states (what should be done), whereas the policy learns what to do in order to get to a goal-state (how it should be done). The intuition motivating this design is that the knowledge about how instructions define goals should generalize more strongly than the knowledge about which behavior is needed to execute instructions. Following this intuition, we propose to reuse a reward model trained in AGILE as a reward function for training or fine-tuning policies.

**Relation to GAIL**  AGILE is strongly inspired by—and retains close relations to—Generative Adversarial Imitation Learning (GAIL; Ho & Ermon, 2016), which likewise trains both a reward function and a policy. The former is trained to distinguish between the expert's and the policy's trajectories, while the latter is trained to maximize the modelled reward. GAIL differs from AGILE in a number of important respects. First, AGILE is conditioned on instructions $c$ so a single AGILE agent can learn combinatorially many skills rather than just one. Second, in AGILE the reward model observes only states $s_i$ (either goal states from an expert, or states from the agent acting on the environment) rather than state-action traces $(s_1, a_1), (s_2, a_2), \ldots$, learning to reward the agent based on "what" needs to be done rather than according to "how" it must be done. Finally, in AGILE the policy's reward is the thresholded probability $[D_\phi(c, s_t)]$ as opposed to the log-probability $\log D_\phi(s_t, a_t)$ used in GAIL. Our reasoning for this change is that, when adapted to the setting with goal-specifications, a GAIL-style reward $\log D_\phi(c, s_t)$ could take arbitrarily low values for intermediate states visited by the agent, as the reward model $D_\phi$ becomes confident that those are not goal states. Empirically, we found that dropping the logarithm from GAIL-style rewards is indeed crucial for AGILE's performance, and that using the probability $D_\phi(c, s_t)$ as the reward $\hat{r}_t$ results in a performance level similar to that of the discretized AGILE reward $\hat{r}_t = [D_\phi(c, s_t)]$.

## 3 EXPERIMENTS

We experiment with AGILE in a grid world environment that we call GridLU, short for Grid Language Understanding and after the famous SHRDLU world (Winograd, 1972). GridLU is a fully observable grid world in which the agent can walk around the grid (moving up, down left or right), pick blocks up and drop them at new locations (see Figure 3 for an illustration and Appendix C for a detailed description of the environment).

### 3.1 MODELS

All our models receive the world state as a 56x56 RGB image. With regard to processing the instruction, we will experiment with two kinds of models: Neural Module Networks (NMN) that treat the instruction as a structured expression, and a generic model that takes an unstructured instruction representation and encodes it with an LSTM.

Because the language of our instructions is generated from a simple grammar, we perform most of our experiments using policy and reward model networks that are constructed using the NMN (Andreas et al., 2016) paradigm. NMN is an elegant architecture for grounded language processing in which a tree of neural modules is constructed based on the language input. The visual input is then fed to the leaf modules, which send their outputs to their parent modules, which process is repeated until the root of the tree. We mimic the structure of the instructions when constructing the tree of modules; for example, the NMN corresponding to the instruction $c_1$=*NorthFrom('red', Shape('circle', SCENE)), Color('blue', Shape('square', SCENE)))* performs a computation $h_{NMN} = m_{NorthFrom}(m_{red}(m_{circle}(h_s)), m_{blue}(m_{square}(h_s))))$, where $m_x$ denotes the module corresponding to the token $x$, and $h_s$ is a representation of state $s$. Each module $m_x$ performs a convolution (weights shared by all modules) followed by a token-specific Feature-Wise Linear Modulation (FiLM) (Perez et al., 2017): $m_x(h_l, h_r) = ReLU((1+\gamma_x)\odot(W_m * [h_l; h_r])\oplus\beta_x)$,

where $h_l$ and $h_r$ are module inputs, $\gamma_x$ is a vector of FiLM multipliers, $\beta_x$ are FiLM biases, $\odot$ and $\oplus$ are element-wise multiplication and addition with broadcasting, $*$ denotes convolution. The representation $h_s$ is produced by a convnet. The NMN's output $h_{NMN}$ undergoes max-pooling and is fed through a 1-layer MLP to produce action probabilities or the reward model's output. Note, that while structure-wise our policy and reward model are mostly similar, they do not share parameters.

NMN is an excellent model when the language structure is known, but this may not be the case for natural language. To showcase AGILE's generality we also experiment with a very basic structure-agnostic architecture. We use FiLM to condition a standard convnet on an instruction representation $h_{LSTM}$ produced by an LSTM. The $k$-th layer of the convnet performs a computation $h_k = ReLU((1+\gamma_k) \odot (W_k * h_{k-1}) \oplus \beta_k)$, where $\gamma_k = W_k^\gamma h_{LSTM} + b_k^\gamma$, $\beta_k = W_k^\beta h_{LSTM} + b_k^\beta$. The same procedure as described above for $h_{NMN}$ is used to produce the network outputs using the output $h_5$ of the 5$^{\text{th}}$ layer of the convnet.

In the rest of the paper we will refer to the architectures described above as FiLM-NMN and FiLM-LSTM respectively. FiLM-NMN will be the default model in all experiments unless explicitly specified otherwise. Detailed information about network architectures can be found in Appendix G.

## 3.2 TRAINING DETAILS

For the purpose of training the policy networks both within AGILE, and for our baseline trained from ground-truth reward $r_t$ instead of the modelled reward $\hat{r}_t$, we used the Asynchronous Advantage Actor-Critic (A3C; Mnih et al., 2016). Any alternative training mechanism which uses reward could be used—since the only difference in AGILE is the source of the reward signal, and for any such alternative the appropriate baseline for fair comparison would be that same algorithm applied to train a policy from ground-truth reward. We will refer to the policy trained within AGILE as AGILE-A3C. The A3C's hyperparameters $\gamma$ and $\lambda$ were set to 0.99 and 0 respectively, i.e. we did not use without temporal difference learning for the baseline network. The length of an episode was 30, but we trained the agent on advantage estimation rollouts of length 15. Every experiment was repeated 5 times. We considered an episode to be a success if the final state was a goal state as judged by a task-specific *success criterion*, which we describe for the individual tasks below. We use the success rate (i.e. the percentage of successful episodes) as our main performance metric for the agents. Unless otherwise specified we use the NMN-based policy and reward model in our experiments. Full experimental details can be found in Appendix D.

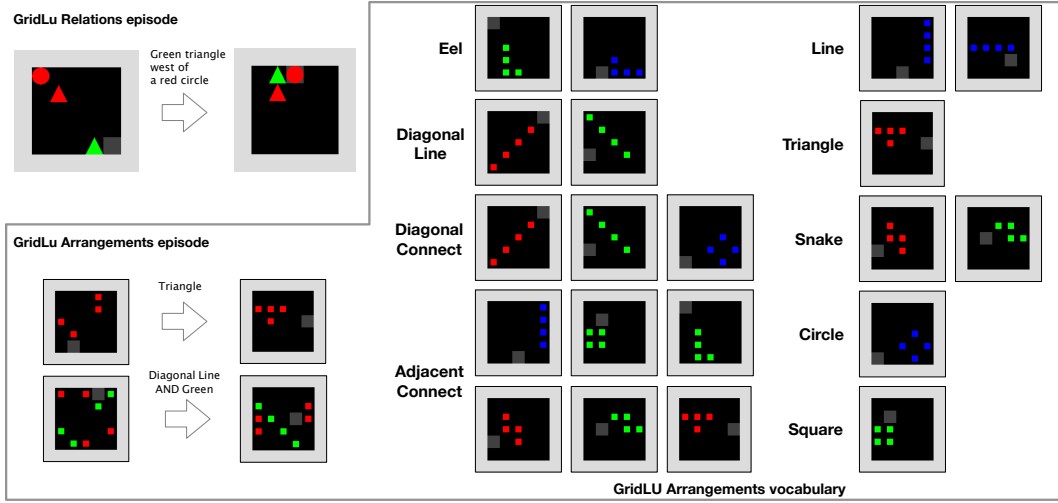

Figure 3: Initial state and goal state for GridLU-Relations (top-left) and GridLU-Arrangements episodes (bottom-left), and the complete GridLU-Arrangements vocabulary (right), each with examples of some possible goal-states.

## 3.3 GRIDLU-RELATIONS

Our first task, GridLU-Relations, is an adaptation of the SHAPES visual question answering dataset (Andreas et al., 2016) in which the blocks can be moved around freely. GridLU-Relations requires the agent to induce the meaning of spatial relations such as *above* or *right of*, and to manipulate the world in order to instantiate these relationships. Named GridLU-Relations, the task involves five spatial relationships (*NorthFrom*, *SouthFrom*, *EastFrom*, *WestFrom*, *SameLocation*), whose arguments can be either the blocks, which are referred to by their shapes and colors, or the agent itself. To generate the full set of possible instructions spanned by these relations and our grid objects, we define a formal grammar that generates strings such as:

$$NorthFrom(Color(`red', Shape(`circle', SCENE)), Color(`blue', Shape(`square', SCENE))) \quad (3)$$

This string carries the meaning 'put a red circle north from (above) a blue square'. In general, when a block is the argument to a relation, it can be referred to by specifying both the shape and the color, like in the example above, or by specifying just one of these attributes. In addition, the *AGENT* constant can be an argument to all relations, in which case the agent itself must move into a particular spatial relation with an object. Figure 3 shows two examples of GridLU-Relations instructions and their respective goal states. There are 990 possible instructions in the GridLU-Relations task, and the number of distinct training instances can be loosely lower-bounded by $1.8 \cdot 10^7$ (see Appendix E for details).

Notice that, even for the highly concrete spatial relationships in the GridLU-Relations language, the instructions are underspecified and somewhat ambiguous—is a block in the top-right corner of the grid *above* a block in the bottom left corner? We therefore decided (arbitrarily) to consider all relations to refer to immediate adjacency (so that Instruction equation 3 is satisfied if and only if there is a red circle in the location *immediately* above a blue square). Notice that the commands are still underspecified in this case (since they refer to the relationship between two entities, not their absolute positions), even if the degree of ambiguity in their meaning is less than in many real-world cases. The policy and reward model trained within AGILE then have to infer this specific sense of what these spatial relations mean from goal-state examples, while the baseline agent is allowed to access our programmed ground-truth reward. The binary ground-truth reward (true if the state is a goal state) is also used as the success criterion for evaluating AGILE.

Having formally defined the semantics of the relationships and programmed a reward function, we compared the performance of an AGILE-A3C agent against a priviliged baseline A3C agent trained using ground-truth reward. Interestingly, we found that AGILE-A3C learned the task more easily than standard A3C (see the respective curves in Figure 4). We hypothesize this is because the modeled rewards are easy to learn at first and become more sparse as the reward model slowly improves. This naturally emerging curriculum expedites learning in the AGILE-A3C when compared to the A3C-trained policy that only receives signal upon reaching a perfect goal state.

We did observe, however, that the A3C algorithm could be improved significantly by applying the auxiliary task of reward prediction (RP; Jaderberg et al., 2016), which was applied to language learning tasks by Hermann et al. (2017) (see the A3C and A3C-RP curves in Figure 4). This objective reinforces the association between instructions and states by having the agent replay the states immediately prior to a non-zero reward and predict whether or not it the reward was positive (i.e. the states match the instruction) or not. This mechanism made a significant difference to the A3C performance, increasing performance to 99.9%. AGILE-A3C also achieved nearly perfect performance (99.5%). We found this to be a very promising result, since within AGILE, we induce the reward function from a limited set of examples.

The best results with AGILE-A3C were obtained using the anticipated negative rate $\rho = 25\%$. When we used larger values of $\rho$ AGILE-A3C training started quicker but after 100-200 million steps the performance started to deteriorate (see AGILE curves in Figure 4), while it remained stable with $\rho = 25\%$.

**Data efficiency**  These results suggest that the AGILE reward model was able to induce a near perfect reward function from a limited set of ⟨instruction, goal-state⟩ pairs. We therefore explored how small this training set of examples could be to achieve reasonable performance. We found that with a training set of only 8000 examples, the AGILE-A3C agent could reach a performance of 60%

(massively above chance). However, the optimal performance was achieved with more than 100,000 examples. The full results are available in Appendix D.

**Generalization to Unseen Instructions**   In the experiments we have reported so far the AGILE agent was trained on all 990 possible GridLU-Relation instructions. In order to test generalization to unseen instructions we held out 10% of the instructions as the test set and used the rest 90% as the training set. Specifically, we restricted the training instances and ⟨instruction, goal-state⟩ pairs to only contain instructions from the training set. The performance of the trained model on the test instructions was the same as on the training set, showing that AGILE did not just memorise the training instructions but learnt a general interpretation of GridLU-Relations instructions.

**AGILE with Structure-Agnostic Models**   We report the results for AGILE with a structure-agnostic FILM-LSTM model in Figure 4 (middle). AGILE with $\rho = 25\%$ achieves a high $97.5\%$ success rate, and notably it trains almost as fast as an RL-RP agent with the same architecture.

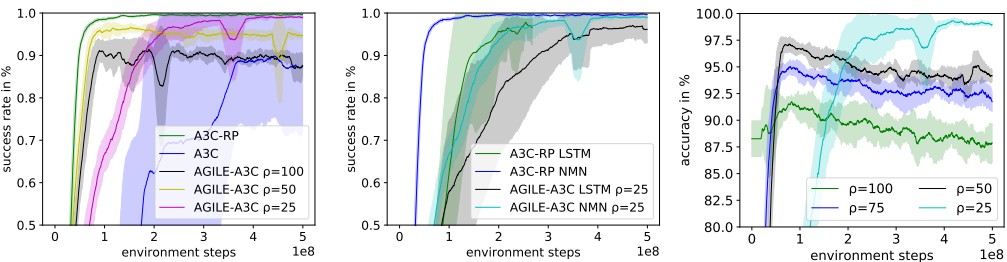

Figure 4: **Left:** learning curves for A3C, A3C-RP (both using ground truth reward), and AGILE-A3C with different values of the anticipated negative rate $\rho$ on the GridLU-Relations task. We report success rate (see Section 3). **Middle:** learning curves for policies trained with ground-truth RL, and within AGILE, with different model architectures. **Right:** the reward model's accuracy for different values of $\rho$.

**Analyzing the reward model**   We compare the binary reward provided by the reward model with the ground-truth from the environment during training on the GridLU-Relation task. With $\rho = 25\%$ the accuracy of the reward model peaks at 99.5%. As shown in Figure 4 (right) the reward model learns faster in the beginning with larger values of $\rho$ but then deteriorates, which confirms our intuition about why $\rho$ is an important hyperparameter and is aligned with the success rate learning curves in Figure 4 (left). We also observe during training that the false negative rate is always kept reasonably low ($<3\%$ of rewards) whereas the reward model will initially be more generous with false positives (20–50% depending on $\rho$ during the first 20M steps of training) and will produce an increasing number of false positives for insufficiently small values of $\rho$ (see plots in Appendix E). We hypothesize that early false positives may facilitate the policy's training by providing it with a sort of curriculum, possibly explaining the improvement over agents trained from ground-truth reward, as shown above.

**The reward model as general reward function**   An instruction-following agent should be able to carry-out known instructions in a range of different contexts, not just settings that match identically the specific setting in which those skills were learned. To test whether the AGILE framework is robust to (semantically-unimportant) changes to the environment dynamics, we first trained the policy and reward model as normal and then modified the effective physics of the world by making all red square objects immovable. In this case, following instructions correctly is still possible in almost all cases, but not all solutions available during training are available at test time. As expected, this change impaired the policy and the agent's success rate on the instructions referring to a red square dropped from $98\%$ to $52\%$. However, after fine-tuning the policy (additional training of the policy on the test episodes using the reward from the previously-trained-then-frozen reward model), the success rate went up to $69.3\%$ (Figure 5). This experiment suggests that the AGILE reward model learns useful and generalisable linguistic knowledge. The knowledge can be applied to help policies adapt in scenarios where the high-level meaning of commands is familiar but the low-level physical dynamics is not.

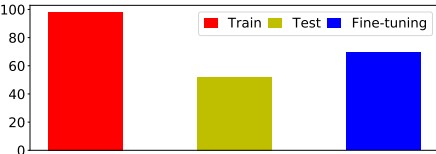

Figure 5: Fine-tuning for an immovable red square.

### 3.4 GRIDLU-ARRANGEMENTS TASK

The experiments thus far demonstrate that even without directly using the reward function AGILE-A3C performs comparably to its pure A3C counter-part. However, the principal motivation for the AGILE framework is to avoid programming the reward function. To model this setting more explicitly, we developed the task **GridLU-Arrangements**, in which each instruction is associated with multiple viable goal-states that share some (more abstract) common form. The complete set of instructions and forms is illustrated in Figure 3. To get training data, we built a generator to produce random instantiations (i.e. any translation, rotation, reflection or color mapping of the illustrated forms) of these goal-state classes, as positive examples for the reward model. In the real world, this process of generating goal-states could be replaced by finding, or having humans annotate, labelled images. In total, there are 36 possible instructions in GridLU-Arrangements, which together refer to a total of 390 million correct goal-states (see Appendix F for details). Despite this enormous space of potentially correct goal-states, we found that for good performance it was necessary to train AGILE on only 100,000 (less than 0.3%) of these goal-states, sampled from the same distribution as observed in the episodes. To replicate the conditions of a potential AGILE application as close as possible, we did not write a reward function for GridLU-Arrangements (even though it would have been theoretically possible), and instead carried out all evaluation manually.

The training regime for GridLU-Arrangements involved two classes of episodes (and instructions). Half of the episodes began with four square blocks (all of the same color), and the agent, in random unique positions, and an instruction sampled uniformly from the list of possible arrangement words. In the other half of the episodes, four square blocks of one color and four square blocks of a different color were initially each positioned randomly. The instruction in these episodes specified one of the two colors together with an arrangement word. We trained policies and reward models using AGILE with 10 different seeds for each level, and selected the best pair based on how well the policy maximised *modelled* reward. We then manually assessed the final state of each of 200 evaluation episodes, using human judgement that the correct shape has been produced as success criterion to evaluate AGILE. We found that the agent made the correct arrangement in 58% of the episodes. The failure cases were almost always in the episodes involving eight blocks[1]. In these cases, the AGILE agent tended towards building the correct arrangement, but was impeded by the randomly positioned non-target-color blocks and could not recover. Nonetheless, these scores, and the compelling behaviour observed in the video (https://www.youtube.com/watch?v=07S-x3MkEoQ), demonstrate the potential of AGILE for teaching agents to execute semantically vague or underspecified instructions.

### 4 RELATED WORK

Learning to follow language instructions has been approached in many different ways, for example by reinforcement learning using a reward function programmed by a system designer. Janner et al. (2017); Oh et al. (2017); Hermann et al. (2017); Chaplot et al. (2018); Denil et al. (2017); Yu et al. (2018) consider instruction-following in 2D or 3D environments and reward the agent for arriving at the correct location or object. Janner et al. (2017) and Misra et al. (2017) train RL agents to produce goal-states given instructions. As discussed, these approaches are constrained by the difficulty of programming language-related reward functions, a task that requires an programming expert, detailed access to the state of the environment and hard choices above how language should map to the world. Agents can be trained to follow instructions using complete demonstrations, that is sequences of correct actions describing instruction execution for given initial states.

---

[1]The agent succeeded on 92% (24%) with 4 (8) blocks.

Chen & Mooney (2011); Artzi & Zettlemoyer (2013) train semantic parsers to produce a formal representation of the query that when fed to a predefined execution model matches exactly the sequence of actions from the demonstration. Andreas & Klein (2015); Mei et al. (2016) sidestep the intermediate formal representation and train a Conditional Random Field (CRF) and a sequence-to-sequence neural model respectively to directly predict the actions from the demonstrations. A underlying assumption behind all these approaches is that the agent and the demonstrator share the same actuation model, which might not always be the case. In the case of navigational instructions the trajectories of the agent and the demonstrators can sometimes be compared without relying on the actions, like e.g. Vogel & Jurafsky (2010), but for other types of instructions such a hard-coded comparison may be infeasible. Tellex et al. (2011) train a log-linear model to map instruction constituents into their groundings, which can be objects, places, state sequences, etc. Their approach requires access to a structured representation of the world environment as well as intermediate supervision for grounding the constituents.

Our work can be categorized as apprenticeship (imitation) learning, which studies learning to perform tasks from demonstrations and feedback. Many approaches to apprenticeship learning are variants of inverse reinforcement learning (IRL), which aims to recover a reward function from expert demonstrations (Abbeel & Ng, 2004; Ziebart et al., 2008). As stated at the end of Section 2, the method most closely related to AGILE is the GAIL algorithm from the IRL family (Ho & Ermon, 2016). There have been earlier attempts to use IRL-style methods for instruction following (MacGlashan et al., 2015; Williams et al., 2018), but unlike AGILE, they relied on the availability of a formal reward specification language. To our knowledge, ours and the concurrent work by Fu et al. (2018) are the first works to showcase learning reward models for instructions from pixels directly. Besides IRL-style approaches, other apprenticeship learning methods involve training a policy (Knox & Stone, 2009; Warnell et al., 2017) or a reward function (Wilson et al., 2012; Christiano et al., 2017) directly from human feedback. Several recent imitation learning works consider using goal-states directly for defining the task (Ganin et al., 2018; Pathak et al., 2018). AGILE differs from these approaches in that goal-states are only used to train the reward module, which we show generalises to new environment configurations or instructions, relative to those seen in the expert data.

## 5 DISCUSSION

We have proposed AGILE, a framework for training instruction-conditional RL agents using rewards from learned reward models, which are jointly trained from data provided by both experts and the agent being trained, rather than reward provided by an instruction interpreter within the environment. This opens up new possibilities for training language-aware agents: in the real world, and even in rich simulated environments (Brodeur et al., 2017; Wu et al., 2018), acquiring such data via human annotation would often be much more viable than defining and implementing reward functions programmatically. Indeed, programming rewards to teach robust and general instruction-following may ultimately be as challenging as writing a program to interpret language directly, an endeavour that is notoriously laborious (Winograd, 1971), and some say, ultimately futile (Winograd, 1972).

As well as a means to learn from a potentially more prevalent form of data, our experiments demonstrate that policies trained in the AGILE framework perform comparably with and can learn as fast as those trained against ground-truth reward and additional auxiliary tasks. Our analysis of the reward model's classifications gives a sense of how this is possible; the false positive decisions that it makes early in the training help the policy to start learning. The fact that AGILEs objective attenuates learning issues due to the sparsity of reward states within episodes in a manner similar to reward prediction suggests that the reward model within AGILE learns some form of shaped reward (Ng et al., 1999), and could serve not only in the cases where a reward function need to be learned in the absence of true reward, but also in cases where environment reward is defined but sparse. As these cases are not the focus of this study, we note this here, but leave such investigation for future work.

As the policy improves, false negatives can cause the reward model accuracy to deteriorate. We determined a simple method to mitigate this, however, leading to robust training that is comparable to RL with reward prediction and unlimited access to a perfect reward function. Another attractive aspect of AGILE is that learning "what should be done" and "how it should be done" is performed by two different model components. Our experiments confirm that the "what" kind of knowledge generalizes better to new environments. When the dynamics of the environment changed at test time,

fine-tuning using frozen reward model allowed to the policy recover some of its original capability in the new setting.

While there is a large gap to be closed between the sort of tasks and language experimented with in this paper and those which might be presented in "real world" situations or more complex environments, our results provide an encouraging first step in this direction. Indeed, it is interesting to consider how AGILE could be applied to more realistic learning settings, for instance involving first-person vision of 3D environments. Two issues would need to be dealt with, namely training the agent to factor out the difference in perspective between the expert data and the agent's observations, and training the agent to ignore its own body parts if they are visible in the observations. Future work could focus on applying third-person imitation learning methods recently proposed by Stadie et al. (2017) learn the aforementioned invariances. Most of our experiments were conducted with a formal language with a known structure, however AGILE also performed very well when we used a structure-agnostic FiLM-LSTM model which processed the instruction as a plain sequence of tokens. This result suggest that in future work AGILE could be used with natural language instructions.

ACKNOWLEDGMENTS

The authors want to thank Serkan Cabi for providing useful feedback. This research was enabled in part by support provided by Compute Canada (www.computecanada.ca).

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

## A  AGILE Pseudocode

---

**Algorithm 1** AGILE Discriminator Training

---

**Require:** The policy network $\pi_\theta$, the discriminator network $D_\phi$, the anticipated negative rate $\rho$, a dataset $\mathcal{D}$, a replay buffer $B$, the batch size $BS$, a stream of training instances $\mathcal{G}$, the episode length $T$, the rollout length $R$.

1: **while** Not Converged **do**
2:    Sample a training instance $(c, s_0) \in \mathcal{G}$.
3:    $t \leftarrow 0$
4:    **while** t ¡ T **do**
5:        Act with $\pi_\theta(c, s)$ and produce a rollout $(c, s_{t...t+R})$.
6:        Add $(c, s)$ pairs from $(c, s_{t...t+R})$ to the replay buffer $B$. Remove old pairs from $B$ if it is overflowing.

7:        Sample a batch $D_+$ of $BS/2$ positive examples from $\mathcal{D}$.
8:        Sample a batch $D_-$ of $BS/(2 \cdot (1 - \rho))$ negative $(c, s)$ examples from $B$.
9:        Compute $\kappa = D_\phi(c, s)$ for all $(c, s) \in D_-$ and reject the top $1 - \rho$ percent of $D_-$ with the highest $\kappa$. The resulting $D_-$ will contain $BS/2$ examples.
10:       Compute $\tilde{L}_D(\phi) = \frac{1}{BS} \sum\limits_{(c,s) \in D_-} -\log(1 - D_\phi(c,s)) + \sum\limits_{(c,g) \in D_+} -\log D_\phi(c_i, g_i)$.
11:       Compute the gradient $\frac{d\tilde{L}_D(\phi)}{d\phi}$ and use it to update $\phi$.
12:       Synchronise $\theta$ and $\phi$ with other workers.
13:       $t \leftarrow t + R$
14:   **end while**
15: **end while**

---

---

**Algorithm 2** AGILE Policy Training

---

**Require:** The policy network $\pi_\theta$, the discriminator network $D_\phi$, a dataset $\mathcal{D}$, a replay buffer $B$, a stream of training instances $\mathcal{G}$, the episode length $T$.

1: **while** Not Converged **do**
2:    Sample a training instance $(c, s_0) \in \mathcal{G}$.
3:    $t \leftarrow 0$
4:    **while** t ¡ T **do**
5:        Act with $\pi_\theta(c, s)$ and produce a rollout $(c, s_{t...t+R})$.
6:        Use the discriminator $D_\phi$ to compute the rewards $r_\tau = [D_\phi(c, s_\tau) > 0.5]$.
7:        Perform an RL update for $\theta$ using the rewards $r_\tau$.
8:        Synchronise $\theta$ and $\phi$ with other workers.
9:        $t \leftarrow t + R$
10:   **end while**
11: **end while**

---

## B  Training Details

We trained the policy $\pi_\theta$ and the discriminator $D_\phi$ concurrently using RMSProp as the optimizer and Asynchronous Advantage Actor-Critic (A3C) (Mnih et al., 2016) as the RL method. A baseline predictor (see Appendix G for details) was trained to predict the discounted return by minimizing the mean square error. The RMSProp hyperparameters were different for $\pi_\theta$ and $D_\phi$, see Table 1. A designated worker was used to train the discriminator (see Algorithm 1). Other workers trained only the policy (see Algorithm 2). We tried having all workers write to the replay buffer $B$ that was used for the discriminator training and found that this gave the same performance as using $(c, s)$ pairs produced by the discriminator worker only. We found it crucial to regularize the discriminator by clipping columns of all weights matrices to have the L2 norm of at most 1. In particular, we multiply incoming weights $w_u$ of each unit $u$ by $\min(1, 1/||w_u||_2)$ after each gradient update as proposed by Srivastava et al. (2014). We linearly rescaled the policy's rewards to the $[0; 0.1]$ interval for both RL and AGILE. When using RL with reward prediction we fetch a batch from the replay buffer and compute the extra gradient for every rollout.

For the exact values of hyperparameters for the GridLU-Relations task we refer the reader to Table 1. The hyperparameters for GridLU-Arrangements were mostly the same, with the exception of the

episode length and the rollout length, which were 45 and 30 respectively. For training the RL baseline for GridLU-Relations we used the same hyperparameter settings as for the AGILE policy.

Table 1: Hyperparameters for the policy and the discriminator for the GridLU-Relations task.

| Group | Hyperparameter | Policy $\pi_\theta$ | Discriminator $D_\phi$ |
|---|---|---|---|
| **RMSProp** | learning rate | 0.0003 | 0.0005 |
| | decay | 0.99 | 0.9 |
| | $\epsilon$ | 0.1 | $10^{-10}$ |
| | grad. norm threshold | 40 | 25 |
| | batch size | 1 | 256 |
| **RL** | rollout length | 15 | — |
| | episode length | 30 | — |
| | discount | 0.99 | — |
| | reward scale | 0.1 | — |
| | baseline cost | 1.0 | — |
| | reward prediction cost (when used) | 1.0 | — |
| | reward prediction batch size | 4 | — |
| | num. workers training $\pi_\theta$ | 15 | 1 |
| **AGILE** | size of replay buffer $B$ | — | 100000 |
| | num. workers training $D_\phi$ | — | 1 |
| **Regularization** | entropy weight $\alpha$ | 0.01 | — |
| | max. column norm | — | 1 |

## C    GridLU Environment

The GridLU world is a $5 \times 5$ gridworld surrounded by walls. The cells of the grid can be occupied by blocks of 3 possible shapes (circle, triangle, and square) and 3 possible colors (red, blue, and green). The grid also contains an agent sprite. The agent may carry a block; when it does so, the agent sprite changes color[2]. When the agent is free, i.e. when it does not carry anything, it is able to enter cells with blocks. A free agent can pick a block in the cell where both are situated. An agent that carries a block cannot enter non-empty cells, but it can instead drop the block that it carries in any non-empty cell. Both picking up and dropping are realized by the INTERACT action. Other available actions are LEFT, RIGHT, UP and DOWN and NOOP. The GridLU agent can be seen as a cursor (and this is also how it is rendered) that can be moved to select a block or a position where the block should be released. Figure 6 illustrates the GridLU world and its dynamics. We render the state of the world as a color image by displaying each cell as an $8 \times 8$ patch[3] and stitching these patches in a $56 \times 56$ image[4]. All neural networks take this image as an input.

## D    Experiment Details

Every experiment was repeated 5 times and the average result is reported.

**RL vs. AGILE**    All agents were trained for $5 \cdot 10^8$ steps.

**Data Efficiency**    We trained AGILE policies with datasets $\mathcal{D}$ of different sizes for $5 \cdot 10^8$ steps. For each policy we report the maximum success rate that it showed in the course of training.

**GridLU-Arrangements**    We trained the agent for 100M time steps, saving checkpoints periodically, and selected the checkpoint that best fooled the discriminator according to the agent's internal reward.

---

[2]We wanted to make sure the that world state is fully observable, hence the agent's carrying state is explicitly color-coded.

[3]The relatively high $8 \times 8$ resolution was necessary to let the network discern the shapes.

[4]The image size is $56 \times 56$ because the walls surrounding the GridLU world are also displayed.

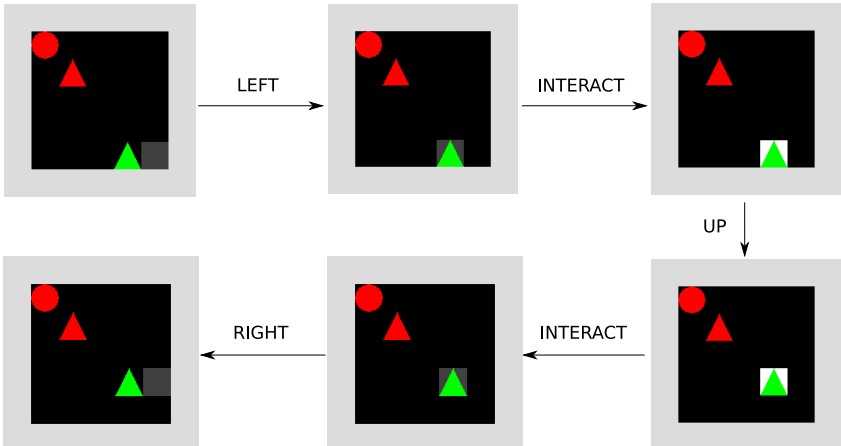

Figure 6: The dynamics of the GridLU world illustrated by a 6-step trajectory. The order of the states is indicated by arrows. The agent's actions are written above arrows.

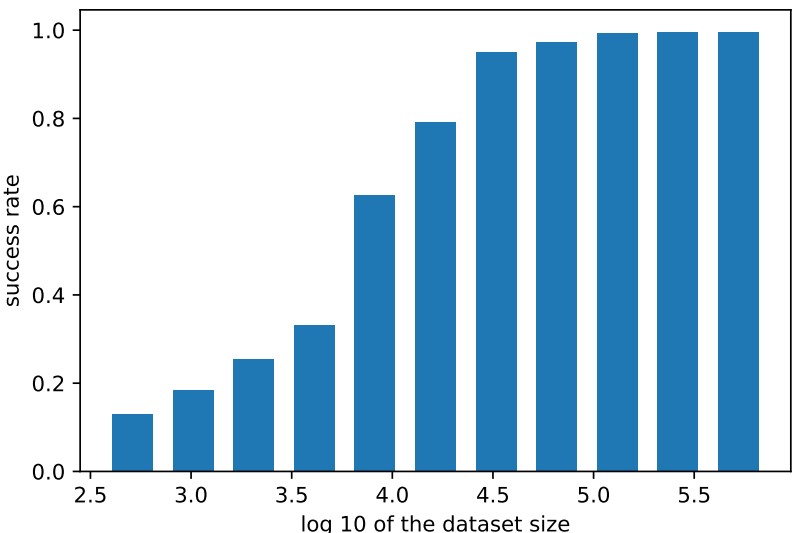

Figure 7: Performance of AGILE for different sizes of the dataset of instructions and goal-states. For each dataset size of we report is the best average success rate over the course of training.

**Data Efficiency** We measure how many examples of instructions and goal-states are required by AGILE in order to understand the semantics of the GridLU-Relations instruction language. The results are reported in Figure 7. The AGILE-trained agent succeeds in more than 50% of cases starting from 8000 examples, but as many as 130000 is required for the best performance.

# E  ANALYSIS OF THE GRIDLU-RELATIONS TASK

## E.1  GRIDLU RELATIONS INSTANCE GENERATOR

All GridLU instructions can be generated from `<instruction>` using the following Backus-Naur form, with one exception: The first expansion of `<obj>` must not be identical to the second expansion of `<obj>` in `<bring_to_instruction>`.

```
<shape> ::= circle | rect | triangle
<color> ::= red | green | blue
```

```
<relation1> ::= NorthFrom | SouthFrom | EastFrom | WestFrom
<relation2> ::= <relation1> | SameLocation

<obj> ::= Color(<color>, <obj_part2>) | Shape(<shape>, SCENE)
<obj_part2> ::= Shape(<shape>, SCENE) | SCENE

<go_to_instruction> ::= <relation2>(AGENT, <obj>) | <relation2>(<obj>, AGENT)
<bring_to_instruction> ::= <relation1>(<obj>, <obj>)
<instruction> ::= <go_to_instruction> | <bring_to_instruction>
```

There are 15 unique possibilities to expand the nonterminal <obj>, so there are 150 unique possibilities to expand <go_to_instruction> and 840 unique possibilities to expand <bring_to_instruction> (not counting the exceptions mentioned above). Hence there are 990 unique instructions in total. However, several syntactically different instructions can be semantically equivalent, such as EastFrom(AGENT, Shape(rect, SCENE)) and WestFrom(Shape(rect, SCENE), AGENT).

Every instruction partially specifies what kind of objects need to be available in the environment. For go-to-instructions we generate one object and for bring-to-instructions we generate two objects according to this partial specification (unspecified shapes or colors are picked uniformly at random). Additionally, we generate one "distractor object". This distractor object is drawn uniformly at random from the 9 possible objects. All of these objects and the agent are each placed uniformly at random into one of 25 cells in the 5x5 grid.

The instance generator does not sample an instruction uniformly at random from a list of all possible instructions. Instead, it generates the environment at the same time as the instruction according to the procedure above. Afterwards we impose two 'sanity checks': are any two objects in the same location or are they all identical? If any of these two checks fail, the instance is discarded and we start over with a new instance.

Because of this rejection sampling technique, go-to-instructions are ultimately generated with approximately 25% probability even though they only represent $\approx 15\%$ of all possible instructions.

The number of different initial arrangements of three objects can be lower-bounded by $\binom{9}{3} = 2300$ if we disregard their permutation. Hence every bring-to-instruction has at least $K = 2300 \cdot 9 \approx 2 \cdot 10^4$ associated initial arrangements. Therefore the total number of task instances can be lower-bounded with $840 \cdot K \approx 1.7 \cdot 10^7$, disregarding the initial position of the agent.

## E.2 DISCRIMINATOR EVALUATION

During the training on GridLU-Relations we compared the predictions of the discriminator with those of the ground-truth reward checker. This allowed us to monitor several performance indicators of the discriminator, see Figure 8.

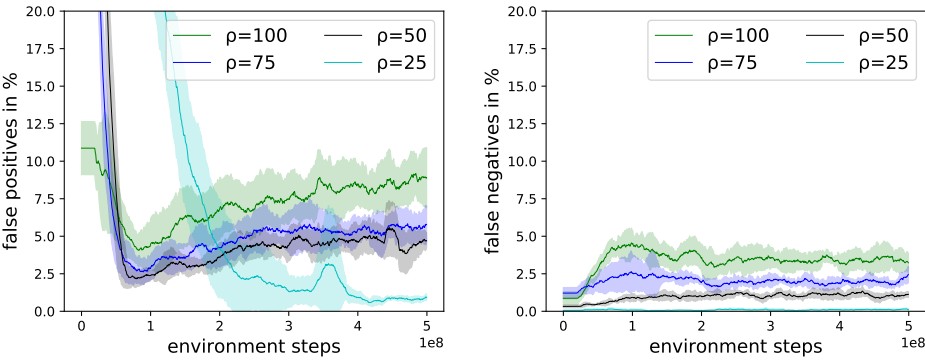

Figure 8: The discriminator's errors in the course of training. **Left:** percentage of false positives. **Right:** percentage of false negatives.

## F ANALYSIS OF THE GRIDLU-ARRANGEMENTS TASK

**Instruction Syntax**   We used two types of instructions in the GridLU-Arrangements task, those referring only to the arrangement and others that also specified the color of the blocks. Examples Connected(AGENT, SCENE) and Snake(AGENT, Color('yellow', SCENE)) illustrate the syntax that we used for both instruction types.

**Number of Distinct Goal-States**   Table 2 presents our computation of the number of distinct goal-states in the GridLU-Arrangements Task.

Table 2: Number of unique goal-states in GridLU-Arrangements task.

| Arrangement | Possible arrangement positions | Possible colors | Possible agent positions | Possible distractor positions | Possible distractor colors | Total goal states |
|---|---|---|---|---|---|---|
| Square | 16 | 3 | 25 | 5985 | 2 | 14,364,000 |
| Line | 40 | 3 | 25 | 5985 | 2 | 35,910,000 |
| Dline | 8 | 3 | 25 | 5985 | 2 | 7,182,000 |
| Triangle | 48 | 3 | 25 | 5985 | 2 | 43,092,000 |
| Circle | 9 | 3 | 25 | 5985 | 2 | 8,079,750 |
| Eel | 48 | 3 | 25 | 5985 | 2 | 43,092,000 |
| Snake | 48 | 3 | 25 | 5985 | 2 | 43,092,000 |
| Connected | 200 | 3 | 25 | 5985 | 2 | 179,550,000 |
| Disconnected | 17 | 3 | 25 | 5985 | 2 | 15,261,750 |
| **Total** | | | | | | **389M** |

## G MODELS

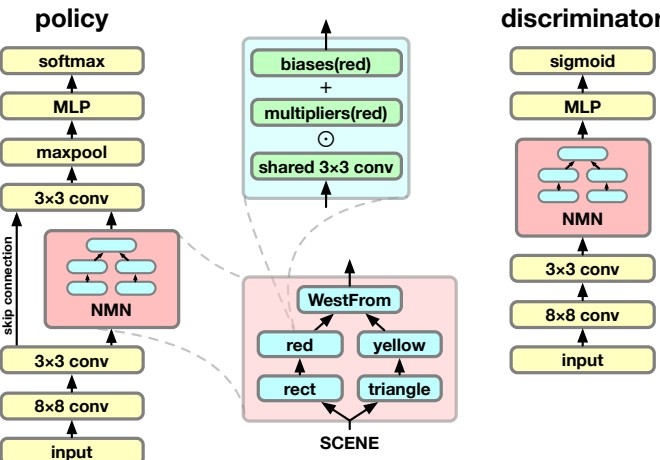

Figure 9: Our policy and discriminator networks with a Neural Module Network (NMN) as the core component. The NMN's structure corresponds to an instruction *WestFrom(Color('red', Shape('rect', SCENE)), Color('yellow', Shape('triangle', SCENE)))*. The modules are depicted as blue rectangles. Subexpressions *Color('red', ...)*, *Shape('rect', ...)*, etc. are depicted as "red" and "rect" to save space. The bottom left of the figure illustrates the computation of a module in our variant of NMN.

In this section we explain in detail the neural architectures that we used in our experiments. We will use $*$ to denote convolution, $\odot$, $\oplus$ to denote element-wise addition of a vector to a 3D tensor with broadcasting (i.e. same vector will be added/multiplied at each location of the feature map). We used ReLU as the nonlinearity in all layers with the exception of LSTM.

**FiLM-NMN**    We will first describe the FiLM-NMN discriminator $D_\phi$. The discriminator takes a 56x56 RGB image $s$ as the representation of the state. The image $s$ is fed through a stem convnet that consisted of an $8x8$ convolution with 16 kernels and a 3x3 convolution with 64 kernels. The resulting tensor $h_{stem}$ had a 5x5x64 shape.

As a Neural Module Metwork (Andreas et al., 2016), the FiLM-NMN is constructed of modules. The module $m_x$ corresponding to a token $x$ takes a left-hand side input $h_l$ and a right-hand side input $h_r$ and performs the following computation with them:

$$m_x(h_l, h_r) = ReLU((1 + \gamma_x) \odot (W_m * [h_l; h_r]) \oplus \beta_x), \tag{4}$$

where $\gamma_x$ and $\beta_x$ are FiLM coefficients (Perez et al., 2017) corresponding to the token $x$, $W_m$ is a weight tensor for a 3x3 convolution with 128 input features and 64 output features. Zero-padding is used to ensure that the output of $m_x$ has the same shape as $h_l$ and $h_r$. The equation above describes a binary module that takes two operands. For the unary modules that received only one input (e.g. $m_{red}$, $m_{square}$) we present the input as $h_l$ and zeroed out $h_r$. This way we are able to use the same set of weights $W_m$ for all modules. We have 12 modules in total, 3 for color words, 3 for shape words, 5 for relations words and one $m_{AGENT}$ module used in go-to instructions. The modules are selected and connected based on the instructions, and the output of the root module is used for further processing. For example, the following computation would be performed for the instruction $c_1 =$ NorthFrom(Color('red', Shape('circle', SCENE)), Color('blue', Shape('square', SCENE))):

$$h_{nmn} = m_{NorthFrom}(m_{red}(m_{circle}(h_{stem})), m_{blue}(m_{square}(h_{stem}))), \tag{5}$$

and the following one for $c_2 =$ NorthFrom(AGENT, Shape('triangle', SCENE)):

$$h_{nmn} = m_{NorthFrom}(m_{AGENT}(h_{stem}), m_{triangle}(h_{stem})). \tag{6}$$

Finally, the output of the discriminator is computed by max-pooling the output of the FiLM-NMN across spatial dimensions and feeding it to an MLP with a hidden layer of 100 units:

$$D(c, s) = \sigma(w^T ReLU(W \text{maxpool}(h_{nmn}) + b)), \tag{7}$$

where $w$, $W$ and $b$ are weights and biases, $\sigma(x) = e^x/(1 + e^x)$ is the sigmoid function.

The policy network $\pi_\phi$ is similar to the discriminator network $D_\theta$. The only difference is that (1) it outputs softmax probabilites for 5 actions instead of one real number (2) we use an additional convolutional layer to combine the output of FiLM-NMN and $h_{stem}$:

$$h_{merge} = ReLU(W_{merge} * [h_{nmn}; h_{stem}] + b_{merge}), \tag{8}$$
$$\pi(c, s) = \text{softmax}(W_2 ReLU(W_1 \text{maxpool}(h_{merge}) + b_1) + b_2), \tag{9}$$

the output $h_{merge}$ of which is further used in the policy network instead of $h_{nmn}$.

Figure 9 illustrates our FiLM-NMN policy and discriminator networks.

**FiLM-LSTM**    For our structure-agnostic models we use an LSTM of 100 hidden units to predict FiLM biases and multipliers for a 5 layer convnet. More specifically, let $h_{LSTM}$ be the final state of the LSTM after it consumes the instruction $c$. We compute the FiLM coefficients for the layer $k \in [1; 5]$ as follows:

$$\gamma_k = W_k^\gamma h_{LSTM} + b_k^\gamma, \tag{10}$$
$$\beta_k = W_k^\beta h_{LSTM} + b_k^\beta, \tag{11}$$

and use them as described by the equation below:

$$h_k = ReLU((1 + \gamma_k) \odot (W_k * h_{k-1}) \oplus \beta_k), \tag{12}$$

where $W_k$ are the convolutional weights, $h_0$ is set to the pixel-level representation of the world state $s$. The characteristics of the 5 layers were the following: (8x8, 16, VALID), (3x3, 32, VALID), (3x3, 64, SAME), (3x3, 64, SAME), (3x3, 64, SAME), where $(mxm, n_{out}, p)$ stands for a convolutional layer with $mxm$ filters, $n_{out}$ output features, and $p \in \{\text{SAME}, \text{VALID}\}$ padding strategy. Layers with $p = $ VALID do not use padding, whereas in those with $p = $ SAME zero padding is added in order to produce an output with the same shape as the input. The layer 5 is also connected to layer 3 by a residual connection. Similarly to FiLM-NMN, the output $h_5$ of the convnet is max-pooled and fed into an MLP with 100 hidden units to produce the outputs:

$$D(c, s) = \sigma(w^T ReLU(W maxpool(h_5) + b)), \tag{13}$$
$$\pi(c, s) = \text{softmax}(W_2 ReLU(W_1 \text{maxpool}(h_5) + b_1) + b_2). \tag{14}$$

**Baseline prediction**   In all policy networks the baseline predictor is a linear layer that took the same input as the softmax layer. The gradients of the baseline predictor are allowed to propagate through the rest of the network.

**Reward prediction**   We use the result $h_{maxpool}$ of the max-pooling operation (which was a part of all models that we considered) as the input to the reward prediction pathway of our model. $h_{maxpool}$ is fed through a linear layer and softmax to produce probabilities of the reward being positive or zero (the reward is never negative in AGILE).

**Weight Initialization**   We use the standard initialisation methods from the Sonnet library[5]. Bias vectors are initialised with zeros. Weights of fully-connected layers are sampled from a truncated normal distribution with $\sigma = \frac{1}{\sqrt{n_{in}}}$, where $n_{in}$ is the number of input units of the layer. Convolutional weights are sampled from a truncated normal distribution with $\sigma = \frac{1}{\sqrt{fan_{in}}}$, where $fan_{in}$ is the product of kernel width, kernel height and the number of input features.

---

[5]https://github.com/deepmind/sonnet/

