# OpenReview forum: "Learning to Understand Goal Specifications by Modelling Reward"
_ICLR.cc/2019/Conference_

### Official Review · AnonReviewer1 · 2018-10-29
**This is a well-written paper. The idea of learning rewards from instructions is interesting although quite straightforward. The experiments show improvement over baseline reward, but they were performed on simple synthetic tasks.**

**Rating:** 6
**Confidence:** 4

**Review:**

The paper presents an approach for simultaneously learning policies and reward functions for reaching goals that are described by an instruction providing spatial relations among objects. The proposed platform, called Adversarial Goal-Induced Learning from Examples (AGILE), is composed of an off-the-shelf RL module like A3C and a separate module for learning a reward function, implemented using the NMN paradigm. The RL module is trained using the reward function learned by the reward module. The reward module is trained to map a given <instruction, state> into a score between 0 and 1 depending on how well the provided state satisfies the instructions provided in the instruction. The returned score is used as a reward function. The training of the reward function is performed by using a dataset of positive examples, and using the states visited by the agent while it's learning as negative examples. To account for the fact that the agent becomes better over time and its visited states can no longer be used as negative examples, the authors proposed a heuristic where the states visited by the agent are not all used as negative examples, but only those that have the lowest scores.
The paper also presents an empirical evaluation of the proposed approach on a synthetic task where the agent is tasked with move bocks of different shapes and colors to a desired final configuration. The AGILE approach was compared to the baseline A3C algorithm where a sparse binary reward signal was used only whenever the agent reaches the goal state. AGILE is also compared to A3C with an auxiliary task of reward prediction.
The paper is clearly written and technically strong. However, I have two issues with this paper: 1) the proposed approach is a simple combination of A3C and the NMN architecture, 2) the experiments are performed on simple synthetic tasks that make learning spatial relations fairly easy, I would love to see more real images as it has been demonstrated in prior works on learning  spatial relations. It is not clear from these experiments if the proposed approach will scale up to higher-dimensional inputs. Moreover, there are several stability issues that can be caused by the proposed approach. For instance, the reward function is changing over time, how does that affect the learning rate? Also, instead of using the learned policy itself to generate negative examples and run into non IID data, instabilities, and increasingly good negative examples, why not use a fixed dataset of negative examples generated with a random policy? It would be interesting to do perform an experiment where you compare to the classical reward learning setup where you simply provided labeled positive and negative examples and classify them offline, then use the learned reward function online for RL.
How did you tune the hyper-parameter \rho (percentage of negative examples to discard) for specific tasks? Do you have any guarantees for this approach?
In the generalization experiments, it is mentioned that 10% of the instructions are held out. Are these 10% randomized?

---

> ### Author Response · Authors · 2018-11-07
> **Thank you for your comments (part 1)**
>
> We thank Reviewer 1 for their review and statement of support for the paper’s technical contributions. We hope, during this discussion period, to get a better understanding of your concerns and hopefully address them, making clarifications in the paper where needed.
>
> First, could you please clarify your first concern that “the proposed approach is a simple combination of A3C and the NMN architecture”? While this is an accurate portrayal of some of our key results, we stress that:
>
> 1) we also report results for LSTM + FiLM-ConvNet architectures for both policy and reward model networks, to showcase performance within AGILE when the syntax of the language is not given (see “AGILE with Structure-Agnostic Models” in Section 3).
>
> 2) use of A3C for our policy network is not an essential part of AGILE. AGILE is a general framework for jointly training policies and reward models, both conditioned on language instructions. Any RL method and network architecture can be used, since the only difference between “traditional” RL and AGILE is the source of the reward: in the former case, it comes from the environment, and in the latter, from the jointly learned reward model. As such, for both the baseline and the AGILE-based model, the fact that we used A3C is not important; all that is important is that we use the same RL algorithm and architecture for both the baseline and AGILE.
>
> With this in mind, do you still believe this aspect of our evaluation is cause for concern?
>
> Second, you suggest experiments which are more visually realistic with more complex relations. We agree that this is where this research should be going, and discuss such further work in section 5, along with the particular challenges we anticipate it will bring. We would be remiss not to point out two things, with regard to our current experiments:
>
> 1) While simple, the environment and tasks are actually quite diverse and approached without the simplifying assumptions typically seen in grid worlds:
>     a) There are over 1,000 instructions in GridLU-Relations, each specifying a different task with millions of different initial environment states to solve.
>     b) The environment is observed, by both policy and reward model, at the pixel level, without predefined notions of what and where objects and their boundaries are, or “privileged features” indicating which predicates apply to which objects (or even which predicates exist).
>
> 2) As we see in the left plot of Figure 3, the A3C baseline results show that, due to the two points made above, even when the reward is specified by the environment (for our baselines), this is quite a difficult multi-task RL problem for fairly modern agent architectures and RL algorithms.  This is because it may take over a dozen steps to optimally obtain a goal state, the task is changing every episode, and reward is sparse. When you add to that the need to jointly infer the reward function from a limited number of expert examples, this constitutes a fairly significant machine learning problem.
>
> While we agree that there are no guarantees with regard to scalability to more realistic environments “out of the box”, we hope you will agree this makes a substantial contribution by showing that it is possible to obtain agents which align with expert notions of reward through the proxy of learned reward functions, from examples, and that it is reasonable to leave further developments which scale to more complex environments for further research.

---

> ### Author Response · Authors · 2018-11-07
> **Thank you for your comments (part 2)**
>
> Finally, to answer your questions at the end of the review:
>
> 1) The primary stability issues that emerge when training have to do with the reward model decaying when the policy starts performing well, which lead to our false negative elimination method introduced in “Dealing with False Negatives”, which lead to stable training. Despite the “non-stationary” nature of the reward model, there was no need to manually change the learning rate during training (although note that we used RMSProp optimisers for both the policy and reward model, which adjusts the learning rate over time as a function of historical gradient norm).
>
> 2) In AGILE, as is the case for GANs and GAN-inspired methods such as GAIL, drawing the negative examples from the “generative model” (the policy), is crucial for obtaining a discriminator which is tailored to evaluating the policy against the reference goal states. For any non-trivial problem with a reasonably large state space, getting negative examples from a random policy would cover very little of the state space (e.g. it’s unlikely to provide examples of the agent holding an object), so coverage cannot be expected to be very good.
>
> 3) We tuned rho via grid-search, as with other hyperparameters. Whereby each instruction corresponds to a specific task, we used a global rho for the entire task set (e.g. GridLU-Relations). We do not have theoretical guarantees to offer, but empirical study shows that training is more stable for low values of rho (trading off longer training for better final results).
>
> 4) When holding out 10% of the instructions, these were randomly chosen from the space of possible instructions.
>
> Thank you for these questions. We will try to make sure the answers to them are clearer just from reading the paper, in our revised draft.

---

### Official Review · AnonReviewer3 · 2018-10-31
**Interesting Work -- Confusing Contextualization with Prior Work**

**Rating:** 7
**Confidence:** 5

**Review:**


==========
Update
==========

Upon reviewing the paper revision and the author comments to my and the other reviewers' comments, I will revise my suggestion to that of acceptance. As I said in my summary, my primary concern was novelty with respect to prior work which the authors have clarified. They have also increased the rigor of their experimental results by providing variances in the plots.
I think this work will be of interest to the community.


==========
Strengths:
==========

- The problem of learning to predict state rewards given language in interesting and useful.

- The proposed AGILE framework is intuitively simple and works with any existing RL framework.

- With the models and tasks explored in this paper, the approach does seem to learn to evaluate whether a state matches the instructions quite well.

- The writing is very clear and direct.

==========
Concerns:
==========

[A] The discussion of differences to the closely related GAIL methodology is left until the related work after experiments. Given the similarities between GAIL and AGILE, this seems too late. The authors list three major differences between AGILE and GAIL:

1) AGILE is conditioned on a goal specification, language in this case. GAIL is unconditioned and trained for one task.
2) AGILE takes only the final/goal state rather than a trajectory like in GAIL.
3) AGILE discretizes the discriminator probability when assigning reward, GAIL does not.

Some concerns about each:

1) This is an interesting and fair difference but also a necessary and somewhat obvious modification to GAIL in tasks with explicit goal-specification.

2) This does not seem like an improvement, but rather a loss of generality. The authors justify this change saying "in AGILE the reward model observes only states s_i (either goal states from an expert, or states from the agent acting on the environment) rather than traces (s1, a1),(s2, a2), . . ., learning to reward the agent based on “what” needs to be done rather than according to “how” it must be done."

In many real applications, the how is deeply important. For instance, navigation in the world is both a "what" (arrive at location X) and a "how" (in fastest time without hitting anything or in such a way that humans aren't frightened). Further, the trace includes the final state such that the "what" is recoverable in instances where the "how" is unimportant, as in the set of tasks presented in this paper.

3) Letting the paper speak on this subject: "We considered this change of objective necessary because the GAIL-style reward would take arbitrarily low values for intermediate states visited by the agent, as the reward model will be confident as those are not goal states. The binary reward in AGILE carries a clear message to the policy that all non-goal states are equally undesirable." Firstly, all non-goal states are not equally undesirable in that some lead more easily to goal states though it is fair to argue this should be learned by the policy through expected reward. My primary gripe is the footnote following these sentences which says: "We tried values other than 0.5 for the binarization threshold, as well as not binarizing and using Dφ(c, st) directly as the reward. We got similar but slightly worse results." This seems to imply that this difference does not matter significantly, especially if different thresholds received significantly different hyperparameter tuning effort or were not conducted under multiple runs of random seeds.

A pessimistic summary would place AGILE to be a conditional GAIL with reduced ability to represent intermediate or trajectory based rewards and a possibly slightly helpful reward discretization scheme. Don't get me wrong, I think an conditional extension to GAIL is interesting and worth sharing with the community. However, this discussion comes very late and includes a design decisions (2/3) that I find poorly justified in text and completely unjustified experimentally.

I would like to hear from the authors if any of these criticisms are inaccurate. I would also welcome experiments evaluating the effect of these design decisions.

[B] In 3.2 its reported that each experiment was repeated five times however the presented results are not described as means and no variances are shown. I would like to see the results plots with shaded variances from at least 5 runs with differing random seeds.

[C] Unless I'm mistaken, the proposed architecture could also be trained with reward prediction. It would be interested in that case to see if improvement seen between A3C and A3C-AGILE extend to A3C-RP and A3C-RP-AGILE. As the authors note, the AGILE framework simply changes the source of the reward and is amicable to any RL approach. I would like to see this comparison.

[D] The reward generalization experiments seemed surprising to me. The policy was fine-tuned on the test environments but only improved from 52% to 69.3%. Trying to think about this more, I'm having trouble disentangling whether this implies poor generalization of the reward function or increased difficulty in policy learning. Could the authors provide the A3C and A3C-RP baselines for this experiment to help clarify?

[E] Just a Curiosity: What exactly is done in L2 weight clipping? (Training details in supplement)

[F] Just a Thought: In the reward-prediction (RP) setting, both the RP model and the policy share parameters. It would be possible with such an architecture to still apply the AGILE loss and I would be curious to see if this leads to interesting changes in performance. I understand that one of the advantages to learning a separate reward model is to generalize to new policies, but it is unclear if this approach would generalize less well (and finding it out would be cool!)

==========
Overview:
==========

I think extending generative adversarial imitation learning to a task-conditional setting a cool step made even more interesting in this work by having the task-specification be in compositional language. Further, the results and analysis are generally interesting though I do note some weaknesses above. Aside from some questions about the experiments, I'm mostly concerned about the positioning of the paper -- specifically with respect to prior work.  I'm looking forward to hearing from the authors and other reviewers.

---

> ### Author Response · Authors · 2018-11-08
> **on the differences between AGILE and GAIL and other comments (part 1)**
>
> We thank reviewer 3 for their very detailed comments, which while critical of the paper, give us much to respond to, and much to consider with regard to improving the paper. We will be honest: if indeed the crucial failing of the paper is its failure to successfully position itself with regard to GAIL, in particular due to the placement of the paragraph discussing comparison, we find that a score of 4 is a little strict. However, we appreciate the reviewer has been very rigorous in explaining their points regarding this weakness. We hope that through discussion we will both be able to present cogent counter-arguments to these objections where applicable, and otherwise satisfy the reviewer that the positioning of the paper can be improved, to their satisfaction, with minor revisions to the paper.
>
> As the reviewer correctly summarized and if we put the conditioning on instructions aside, a key key difference (item 2) of AGILE from GAIL is the fact that AGILE uses images of goal-states as the training data instead of state-action trajectories required for GAIL. We believe that this difference does make AGILE applicable to situations where GAIL would not be. For example, consider the case when the expert that provides demonstrations has effectors different from that of the trained agent (image a human teaching their household robot to arrange objects on a table: the robot won’t necessarily have 5 fingers). Besides, methods such as GAIL force the learner to imitate the training trajectories in their entirety, which can be a limitation when training trajectories are suboptimal (the above example with the human is applicable again, human may not use the shortest path to the goal when they perform their manipulations). Relying on goal-states only brings a certain extra flexibility that GAIL-like methods do not provide, admittedly, at the cost of restricting the applicability of AGILE to declarative instructions, i.e. those which can be verified by the final state only. We believe, however, that such instructions make a common and frequent case in instruction-following setups, and methods that are optimized for this case (such as AGILE ) are worth investigating.
>
> Reviewer has also requested a clarification with regard to the difference between rewards provided by the discriminator in GAIL and AGILE. We admit that the paper may have been not clear enough on this point, as it stressed the discretization of the reward, whereas the real key difference is the switch from log D(...) as the reward in GAIL to (possibly discretized) D(...) in AGILE. The discriminator is trained to output arbitrary low values of log D(c, s) in AGILE for the non-goal-states, and using log D(c, s) as the reward would arbitrarily punish the policy for entering intermediate states that are clearly not goal-states but yet may be useful in the near future. We are currently performing an extra experiment to validate this claim, in which we use log D(c, s) as the reward, and so far we do not see the job getting above 70% success rate, confirming our intuition that GAIL-style log D(c, s) is not an appropriate reward for AGILE. We will update you on the progress of this additional investigation.
>
> Notwithstanding the above differences with GAIL, it’s true that GAIL and AGILE both can be broadly characterised as inverse RL methods, and as such AGILE could be presented as “just” extending IRL to the instruction-conditional case (and we believe we are reasonably up-front about this in the introduction). With all due respect, it is unfair to characterise this observation as a concern, or qualify it as an “obvious extension”: it is, to our knowledge, one of the first substantial studies of instruction-conditional IRL over pixel observations. Experiments show generalisation of the reward model to both new observed states and held-out instructions. The model and objective design underpinning these results are, while inspired by GAIL and related techniques, original insofar as new methods (e.g. discriminator rejection) had to be incorporated to make it work. We think that, on the strength of extending IRL techniques to multi-task contexts, where the task is specified by language, is new enough to warrant publication, and hope you will agree.
>
> (see part 2 for continuation)

---

> ### Author Response · Authors · 2018-11-08
> **on the differences between AGILE and GAIL and other comments (part 2)**
>
> (see part 1 for the beginning of the response)
>
> Bearing in mind our main argument that GAIL and AGILE have fundamental differences which can and will be presented in more detail in the revision we are drafting, with the help of your comments and questions, we hope that we have addressed the core concern that “AGILE to be a conditional [IRL, rather than GAIL] ] with reduced ability to represent intermediate or trajectory based rewards and a possibly slightly helpful reward discretization scheme” in that it presents a novel approach to learning inherently multi-task (language-conditional) policies from expert data, provides a set of experiments to prove the concept thereof, and while differentiating itself from related work, is reasonably clear about its self-imposed limitations and about where future research can make improvements.
>
> We conclude our response to by replying to the rest of your comments:
>
> [B] Thank you for the suggestion to plot shaded variances, we will do so and update our submission as soon as possible.
>
> [C] In our early investigations the reward prediction objective did not help AGILE as much as it helped A3C. We find it unsurprising as reward prediction is especially necessary in cases when the reward is sparse, and AGILE’s reward is more dense than the groundtruth one. This may also have something to do with the fact that the AGILE rewards are constantly changing, and hence predicting them may be not as valuable of an objective.
>
> [D] “We thank the reviewer for posing this very interesting question. We trained the A3C-RP baseline with immovable red square in our preliminary investigations and it also did not achieve a perfect performance. We believe that learning the policy that can deal with the remaining corner cases may be the bottleneck in this case, however, we will rerun this experiment and update you on the exact numbers.“
>
> [E] The L2 weight clipping makes sure that incoming weights of a neuron have a total L2 norm of at most C, where C is a hyperparameter. This regularization is a hard alternative to weight decay proposed (to the best of our knowledge) in https://arxiv.org/pdf/1207.0580.pdf . We will make sure to add a description of this method in Appendix.
>
> [F] Thank for this interesting experiment suggestion. Sharing weights between the policy and the discriminator is indeed an option, which we have not tried mostly because of the stability concerns. We will definitely consider this in our future work.

---

> > ### Comment · AnonReviewer3 · 2018-11-26
> > **A Late Response**
> >
> > I thank the authors for their replies to reviewer comments throughout. I've read them as they've come in but have to apologize for my slowness in replying.
> >
> > Re: "We will be honest: if indeed the crucial failing of the paper is its failure to successfully position itself with regard to GAIL, in particular due to the placement of the paragraph discussing comparison, we find that a score of 4 is a little strict."
> >
> > As we are speaking honestly (though I hope we need not differentiate between doing so and not), given what I take to be strong similarities between AGILE and GAIL, it's discussion being presented so late and without much fanfare left me as a reader with the lingering sense of being duped! There are strong ties between the two methods both in terms of mechanics and motivation that should be discussed -- if for no other reason than to allow readers new to the area a useful notion of heredity. While a rating of 4 is a bit strict, it was given as a worst case should you folks not respond. As I said in my summary, I did in fact look forward to hearing from you :)
> >
> > And hear from you I have! Thanks for the thorough response to my points / questions.
> > I'll respond to a few below:
> >
> > Re: Trajectories vs. End States
> >
> > I appreciate the argument for the flexibility of end-state specification. I will point out that as far as I understand, in both AGILE and GAIL the intermediate trajectory is rewarded according to the trained discriminator -- the key difference being GAIL's reward predictor is trained on trajectories whereas AGILE's is trained on goal state / condition pairs. I think a clearer discussion of the limitations (and flexibility) this choice provides would be very useful. Further, experimental comparison with and without full trajectory training of the discriminator would be very useful.
> >
> >
> > Re: Form of Reward
> >
> > Thank you for the clarification. I did miss the log being dropped. With a refocusing of this section on this difference rather than the discretization, this should be fine. Doubly fine if an appendix were to include ablations of these choices.
> >
> >
> > Re: Extension to Instruction-Conditional IRL
> >
> > As I said in my review, this is an interesting and fair difference and as I echo in my summary an interesting part of this work. I do maintain it is somewhat an obvious extension though that does not limit its usefulness. Listing it under a concern was thoughtless of me.
> >
> >
> > Re: Variances
> >
> > Thanks for adding the variances! Given the high variance of RL methods in general, adding these is a significant help to the community! Looking over these, the claims of the paper still hold well.
> >
> >
> > I will revise my review rating to reflect my increased confidence in this submission. As a personal note, I would like to thank you for your detailed responses to my concerns and those of my much-more-talkative fellow reviewers. The discussion as a whole has been valuable.

---

> > > ### Author Response · Authors · 2018-11-28
> > > **Final Remarks**
> > >
> > > We thank Reviewer 3 for taking the time to consider our response and giving us a detailed explanation of why our current presentation of differences between AGILE and GAIL may be confusing. In the camera-ready version we will make sure to move the explanation of these differences from Section 4 “Related Work” to Section 2 “Adversarial Goal-Induced Learning from Examples”. The results of the experiments with log D(...) as the reward will also be included in the paper, either in the main text or in the appendix.

---

> ### Author Response · Authors · 2018-11-26
> **final response to Reviewer 3**
>
> We would like to report results of several additional experiments that we have run based on the valuable suggestions by Reviewer 3 (R3).
>
> - We have tried to use GAIL-style rewards, r_t = log D(c, s_t). We found that such a modification of AGILE would not perform better than 70% success rate, much as expected. The policy’s return \sum_t log D(c, s_t) would keep decreasing all the time, as the discriminator got confident in rejecting goal states and outputted very low values of log D(c, s_t), in line with our intuition that using GAIL rewards results in punishing the discriminator arbitrarily for entering intermediate states. We would be happy to incorporate these extra results in the paper should R3 find it necessary.
> - We have compared results of a fine-tuned AGILE policy on the immovable red square task with those of RL with ground truth rewards. We used the soon-to-be-open-sourced PPO-based reimplementation. In this reimplementation fine-tuning PPO-AGILE has improved the performance from 65% to 82%, which is pretty close to the 86% success rate of the PPO baseline. From this we conclude that imperfect 82% performance of fine-tuned PPO-AGILE is mostly due to the increased difficulty in policy learning, and not poor generalization of the reward model.
>
> We hope that R3 finds these extra results informative. We believe our prior arguments about differences between AGILE and GAIL together with these extra results address most of R3’s concerns, and we hope that they considering revisiting their evaluation of the paper.
>
> Besides, the extra results presented above, we have uploaded a revised PDF with the following changes:
> - standard deviations are now displayed as shades in all plots
> - weight norm clipping is explained better in Appendix and a reference to the paper where this method was proposed is added

---

### Official Review · AnonReviewer2 · 2018-11-01
**Well-motivated and innovative idea to resolve the reward-absence problem due to ambiguity, variability, and underspecification in natural language instructed RL.**

**Rating:** 7
**Confidence:** 4

**Review:**

The previous version of the paper was not clear enough in the motivation and uniqueness of the work. After a long and devoted discussion with the authors, we agreed on certain ways of improving the paper presentation, including connection to some related work.

The current paper is much better, so I would like to raise my score to 6. My revised review is:

[orginality and significance]

+ The paper deals with a challenging navigation problem where natural language instructions can be underspecified and the environment is complex---thus a correct reward function being extremely hard to craft.
+ The paper proposed to use a <instruction, state> discriminator D to compute a pseudo reward at each step, which is then used to reinforce an agent in natural-language-guided navigation task. The paper proposed to train the discriminator in an adversarial way---with expert supervised data. The idea is neat, and its effectiveness is empirically supported by extensive experimental results.

[clarity]

+ The paper is well-written. The method is introduced with clear textual description, rigorous math formulations, and good illustration (Figure-1 and -2). The experiments are also well-documented, including training and testing details, results and analysis.

[quality]

+ The paper was not clear at certain points but the authors had helpful discussions with me and the paper was revised accordingly.
+ The experiments were done with multiple random seeds, so I believe the results are convincing. The authors did not only show the numerical results but also shared qualitative videos through anonymous URL.  Overall, it is a good paper.

++++++++++++++++++++++++++++++++++++++++++++++++++++++++++++++++++++
Below is my original review
++++++++++++++++++++++++++++++++++++++++++++++++++++++++++++++++++++

[PROS]

[originality and significance]

The paper proposed to use a <instruction, state> discriminator D to compute the reward at each step, which is then used to reinforce an agent in natural-language-guided navigation task. The paper proposed to train the discriminator in adversarial way. The idea is neat, and its effectiveness is empirically supported by extensive experimental results.

[clarity]

The paper is well-written. The method is introduced with clear textual description, rigorous math formulations, and good illustration (Figure-1). The experiments are also well-documented, including training and testing details, results and analysis. The experiments were done with multiple random seeds, so I believe the results are convincing. The authors did not only show the numerical results but also shared qualitative videos through anonymous URL.  Overall, it is a good paper.

[CONS]

[quality]

The major issue of this paper is the lack of connection to existing related work in the field of dealing with reward sparsity problem. This is a long-standing problem in RL (very common in, but not only restricted to, navigation tasks) and people have proposed reward shaping techniques to handle it. But the paper did not discuss any work in this direction. For references, please first check this seminal work and then follow the line of research:

Ng, Andrew Y and Harada, Daishi and Russell, Stuart, ICML 1999, Policy invariance under reward transformations: Theory and application to reward shaping

The method proposed in this paper seems a way of automatically shaping the reward, but loses the optimal policy invariance (for how this invariance is ensured in reward shaping, please check out this tutorial: http://www-users.cs.york.ac.uk/~devlin/presentations/pbrs-tut.pdf).

The proposed method has two key components: 1) the discriminator D; and 2) the adversarial training. The method is shown effective in experiments and outperforms appropriate baselines with actual reward. But the design of D and how it is used as reward function seems somewhat ad-hoc.

D is only trained on the final states of episodes (please correct me if I am wrong), but is used at all the steps as part of reward function to determine the stepwise reward, which seems odd. The authors should discuss what (implicit) assumptions they are relying upon to make this method work in this way. The transformation function from D to reward value seems ad-hoc---e.g. why 0.5, why indicator function instead of others (e.g. scaling of indicator function), how it is generalized to non-1/0 (but still sparse) reward cases, etc? Is the method only designed for 1/0-reward cases? The authors should clearly specify if it is the case.

Moreover, the paper compared to RP (Jaderberg 2016), which still reinforces the agent with actual reward but only *shapes the features of the agent* by multi-tasking on predicting the reward of next step (please correct me if this is wrong). Interestingly, the RP method achieves better performance than the proposed method, although it does not address the reward sparsity problem. Could the authors provide any insight about why this happened? Is there any trade-off between these two methods? Is there any setting, in the authors’ opinion, where the proposed method should outperform RP?

[SUMMARY]

I think this is good work---neat idea, nice results and clear writing. But there are indeed some issues that I hope the authors could address. So I gave a score of 5.

---

> ### Author Response · Authors · 2018-11-06
> **the paper is not about reward sparsity, but about learning the reward**
>
> We thank Reviewer 2 for their kind words and detailed review, and for clearly stating what they believe is the limitation of the paper they would like to see addressed. With all due respect, we would be happy to discuss the relation of this work to the sparse reward problem, and to that of learning shaped rewards, but we believe this recommendation stems from a misunderstanding which we hope to clarify through this discussion period.
>
> Simply put, the primary use case of frameworks like AGILE specifically, and of reward modelling / inverse reinforcement learning in general, is where there is *no* reward function implemented (or even obtainable), rather than a sparse one. In such cases, we must learn a reward function from expert-provided information (trajectories, goal states). In contrast, from our understanding of reward shaping in the context of the reward sparsity problem, the ground truth reward is accessible in order to update the policy (or Q/V functions), and a shaped reward function is learned or defined to give a more dense “fake” reward (while preserving optimality guarantees) and make credit assignment easier.
>
> We understand that the source of this confusion may come from the fact that in one of our experiments (GridLU-Relations), the ground truth reward *is* implemented and *is* sparse, but this is only for the purpose of automated evaluation and the training of baselines for comparison. When training agents with AGILE, or in the other experiments (GridLU-Arrangements) there is no reward provided from the environment.
>
> Perhaps we have misunderstood the point the reviewer is making, in which case we hope they can clarify how reward shaping and the sparse reward problem relate to the no-reward scenario in which our work and other work in inverse RL is situated. However, on the assumption that the reviewer has misunderstood the motivation and problem setting for our approach, we would be grateful if they could re-evaluate their assessment in this light, and/or perhaps let us know where would could have been clearer so as to not potentially confuse future readers along similar lines.

---

> > ### Comment · AnonReviewer2 · 2018-11-07
> > **Thanks for nice clarification. Let's clarify ground-truth reward vs. reward functions.**
> >
> >
> > Thanks to the authors for such a clear and detailed clarification---it indeed helps.
> >
> > But before I make a decision on re-evaluation, we need to further clarify something important.
> > I think the most misleading thing is: every time a *reward* is mentioned in the paper, it means different things---sometimes *ground truth reward* but sometimes *reward functions*.
> >
> > Let's first make a clear distinction between them (even though in many other cases we do not have to, I think it is particularly necessary in this paper). Ground truth reward (term borrowed from your paper and rebuttal) is what we eventually evaluate your RL agent on---so no matter how sparse it is, it always exists (I will shortly claim what ground truth reward I think is in your cases, including GridLU-Arrangements); reward function is what we use to REINFORCE the RL agent---it sometimes is the ground-truth (esp. in cases where the ground-truth reward is rich and dense), but sometimes has to be specified (esp. in cases of sparsity.).
> >
> > We describe your settings differently---I call it *sparse reward* and you call it *no-reward*. I think they are both correct---but in different senses. The ground truth reward is indeed 1-0 (depending on if goal state is reached) in all your cases because you eventually evaluate the RL agent on success rate---it is implicit, but it exists! (In more details, when you write ``the agent made the correct arrangement in 58% of the episodes’’, you indeed implicitly assign 1 to the correct arrangement but 0 to others, right?) But in your settings, there is *no (good) reward functions (except the ground-truth one)* available, and you do not want to manually craft one---then you design a method to automatically learn a pseudo/surrogate/fake-reward function. TLDR: Sparse ground truth, no (other) reward function. Does it make sense to you?
> >
> > The connection to reward-shaping might have been more obvious if we agree on the points above: 1) both reward-shaping and your method does not use the ground-truth reward as the reward function; 2) both reward-shaping and your method early-reward agents for reaching some states that are closer (in any appropriate sense) to the goal states. To elaborate 2), I quote Ng et al 1999 here: ``to encourage moving towards a goal, a shaping-reward function that one might choose is F(s,a,s’)=r whenever s’ is closer (in whatever appropriate sense) to the goal than s, and F(s,a,s’)=0 otherwise, where r is some positive reward’’. Does it sound similar to your definition of \hat{r} on page-2? In more details, you learn a neural function to identify goal-state, which will turn out > 0.5 when the current state (representation) is close/similar enough to the goal state, so you give RL agent a positive reward at this step. Is this argument right?
> >
> > There is indeed a difference between (potential-based) reward-shaping and your method. You may lose the optimality (which means: using the pseudo/surrogate/fake-reward function, the optimal policy you obtain will still give you maximal ground-truth 1-0 reward, i.e. your success rate)---many guarantees are missing when people use flexible neural components, so I am not criticizing it. Your method empirically works well in practice---that is good. But the connection needs to be carefully discussed. Or maybe you can even carefully craft your learned reward function such that your method can preserve optimality---you do not have to do this in this paper and rebuttal, but it might be future direction.
> >
> > Does this response clarify my points?
> >
> > After all, your rebuttal is indeed helpful, because it (indirectly though) clarifies my concern of this paragraph ``D is only trained on the final states of episodes … only designed for 1/0-reward cases?’’ Let me answer my own question here: the authors consider the (language-instructed-)navigation-type tasks where the RL agent learns to achieve goal states (in whatever sense). In such cases, the equivalent ground truth reward is always 1-0, so the way of pseudo-reward being related to goal-state-discriminator seems general enough. The goal-state-discriminator is only trained on final states, but can be deployed on all states, because its job is to find the states ``that are close enough to the goal’’.
> >
> > In the end, as I claim in my review, the work is well-motivated and neat. But it needs clarification on some seemingly subtle but important points, and it should be more tightly connected to past related work.

---

> > > ### Author Response · Authors · 2018-11-08
> > > **Additional clarification (part 1)**
> > >
> > > We thank you for your rapid response, and appreciate your being so willing to engage in the discussion. We say this with the deepest respect: there seems to be some fairly fundamental misunderstanding of AGILE (and reward modelling in general) at play here, but we are ready to accept this my in part be due to how we have explained things. We would like, here, to help clarify this misunderstanding, not only with the intent of convincing you that the comparison to related literature made in the paper is fair and (to the extent it can be with the space allowed) fairly complete, but also with the intent of tweaking the description of the method where necessary so that other readers may not arrive at the same conclusions as you.
> > >
> > > Before responding to your latest comments, allow us to state what we understand to be the “sparse reward problem”: in some settings/tasks/environments, the set of states within an episode where the agent receive a scalar reward signal is a very small proportion of the total states experienced (e.g. just the last state). This makes training an agent difficult, since a particularly complex credit assignment problem, with possibly long-range and structured dependencies between rewards and actions, needs to be solved. To this end, techniques such as reward shaping or reward prediction have proposed to give a denser “signal” based on which the agent can effectively learn to maximise the expected reward, in spite of the sparsity of the “true” reward signal.
> > >
> > > Now consider the cases where there is, quite literally, no reward function implemented in the simulator or environment. This means you *cannot* normally train an RL agent. Of course, a human *could* observe an agent operating in such environments, and assign reward but this fairly obviously would not scale to the requirements of any non-trivial agent and environment (even for classical tabular RL approaches). So in that sense, the IRL literature, GAIL, or the present method (AGILE) are not typically seen as addressing a sparse reward problem, but rather exploiting a small amount of expert data in order to extrapolate the reward function that an expert could conceptually have provided to the environment, but didn’t because it was either impossible (because there is no formalisable verification function over the desired task due to ambiguity or underspecification), or intractable (because implementing such a function would be too onerous). Finally, and perhaps most crucially with regard to the orthogonality between no-reward scenarios and sparse-reward scenarios: the reward provided by a reward model in a no-reward scenario might itself be sparse (and therefore require reward shaping techniques to be applied for the agent to learn a policy).

---

> > > ### Author Response · Authors · 2018-11-08
> > > **Additional clarification (part 2)**
> > >
> > > Now, let us turn specifically to your comments.
> > >
> > > > every time a *reward* is mentioned in the paper, it means different things---sometimes *ground truth reward* but sometimes *reward functions*.
> > >
> > > Reward functions are functions which take states (and, in our case, also instructions) and return a scalar reward for the associated timestep where the state was observed. In an environment like Atari games or Go, it is defined within the game logic or the rules of the board game. In “no-reward” scenarios, it has not been implemented, and must be learned by exploiting some auxiliary data (trajectories for GAIL, human preferences in papers like arXiv:1706.03741, or <instruction, goal state example> pairs in AGILE). In contrast, “reward” means “whatever scalar value we are maximising the expected value thereof by changing the agent’s parameters”. It is a scalar value (possibly just 0) provided to the agent at each timestep. The RL algorithm we use to optimise the agent for both the RL baseline (which uses environment reward in experiments where it is available) and the agent in AGILE (which does not, and therefore is applicable in environments/tasks where no reward function exists) is agnostic to the source of the reward. We discuss this at the end of the 2nd paragraph of section 2 (“We note that Equation 1 differs from a traditional RL [...] the reward model to the environment.”).
> > >
> > > > Ground truth reward (term borrowed from your paper and rebuttal) is what we eventually evaluate your RL agent on
> > >
> > > This is perhaps where the source of confusion comes from. In order to compare policies trained using AGILE (a framework meant to be used when there is *no reward* provided by the environment during training) to policies trained under the “idealised” case where an exact reward function exists in the environment (and provided reward), we implemented, in one of our tasks (GridLU-Relations) a reward function, but only use it to train the A3C and A3C-RP baselines, and pretend that it does not exist when training with AGILE. You are correct that we used this “ground-truth” reward function to automate evaluation in our experiments, but this is purely for convenience: this evaluation does not in anyway play a role in the updating of agent parameters, reward model parameters, or any other part of AGILE. For all intents and purposes, we could have done this evaluation manually, using a human, (as was done for AGILE-Arrangements) and the results would be exactly the same.
> > >
> > > > it always exists (I will shortly claim what ground truth reward I think is in your cases, including GridLU-Arrangements)
> > >
> > > This is true of any environment: the objective reward “exists” in that, for a particular task, a human (or other expert) could observe the environments and provide judgements which, if sufficient in number, could be used to train an agent against an RL objective. As we discussed when contrasting “no reward” scenarios to “sparse reward” scenarios, the possibility of occasionally soliciting reward signal from humans and directly optimising the expected value is broadly intractable for all but the most trivial tasks, environments, and agents. When we talk about reward “not existing”, we simply mean that there is no reward function implemented as a program in the environment which would provide reward signal (sparse or otherwise) which would permit tractable optimisation of an agent.

---

> > > > ### Comment · AnonReviewer2 · 2018-11-08
> > > > **ground truth reward --> ground truth reward function; reward function --> learned/specified reward function (that is used for training)**
> > > >
> > > > See my reply to part 3 for details.

---

> > > ### Author Response · Authors · 2018-11-08
> > > **Additional clarification (part 3)**
> > >
> > > > reward function is what we use to REINFORCE the RL agent---it sometimes is the ground-truth (esp. in cases where the ground-truth reward is rich and dense), but sometimes has to be specified (esp. in cases of sparsity.)
> > >
> > > *Reward* is what we use to optimise agents using RL algorithms. As stated previously/above, RL algorithms are intrinsically agnostic to where this reward comes from. When a reward function is implemented within the environment, the reward it provides can be used to train an agent. AGILE, GAIL, and other IRL methods are all about *learning* this reward function from auxiliary data, in the absence of a reward function implemented in the environment.
> > >
> > > > The ground truth reward is indeed 1-0 (depending on if goal state is reached) in all your cases because you eventually evaluate the RL agent on success rate---it is implicit, but it exists!
> > >
> > > We refer you to our comments about the experimental set up above. The fact that a reward function exists in GridLU-Relations and is used for automated evaluation and the training of traditional RL baselines has not bearing on the nature of AGILE (or any other reward modelling approach). The reward signal it provides is not used, and for all intents and purposes does not exist. It’s sole purpose is to compare methods which do not rely on the provision of an environmental reward function to those which do, with the clear caveat that the latter are not applicable in as wide a selection of settings the former.
> > >
> > > > (In more details, when you write ``the agent made the correct arrangement in 58% of the episodes’’, you indeed implicitly assign 1 to the correct arrangement but 0 to others, right?)
> > >
> > > At test time, we explicitly and manually assign 1 to the correct arrangement and 0 to others. This is only for evaluation. The parameters of the model are not updated on the basis of these runs. To be clear: the policy networks require millions of episodes to learn a decent policy (as is the case across DeepRL) and it is not realistic to solicit human evaluation for each of them.
> > >
> > > > But in your settings, there is *no (good) reward functions (except the ground-truth one)* available, and you do not want to manually craft one
> > >
> > > With the risk of repeating ourselves with regard to the experimental protocol, for AGILE there is no ground truth reward function. Only for the purpose of evaluating “traditional RL” baselines, for comparison of the final policies (not the parameter inference procedure!), is there a “ground truth” reward function, but it is not used by or exposed to the reward model or policy within AGILE. As an aside, the need for reward models is also motivated by the case where you *can’t* write a reward function (e.g. imagine formally verifying the completion of ambiguous or underspecified instructions like “clean the room” or “set up the table” against a rich environment), not just the case where you don’t want to.
> > >
> > > > you design a method to automatically learn a pseudo/surrogate/fake-reward function.
> > >
> > > Think of it this way: we want agents to maximise the reward a human would give them if they were observing the agent all the time during training (let’s assume this is impossible, because the agent needs millions of episodes). Typically, reward functions are implemented by humans to “simulate” having this human present during training: they are designed to align with what human judgements would be. Now, if this reward function hasn’t been implemented, what can we do? IRL methods such as GAIL, AGILE, etc are techniques which, based on some evidence from an expert, attempt to “reverse engineer” what the reward function expressing alignment with human values would be. So it’s not about learning a “pseudo/surrogate/fake-reward function” so much as it is about trying to learn what the “true” reward function could be.
> > >
> > > > TLDR: Sparse ground truth, no (other) reward function. Does it make sense to you?
> > >
> > > Unfortunately not, as our experiments for AGILE are about “*No* ground truth, no (other) reward function” (except the one which we are learning).

---

> > > > ### Comment · AnonReviewer2 · 2018-11-08
> > > > **Let's precisely locate where the confusion comes from---no known ground truth reward function (except human annotation).**
> > > >
> > > >
> > > > Thanks for the long and detailed response.
> > > >
> > > > I think we are now converging on some point and let's get it clear here.
> > > > After carefully reading all your replies (part1---4), I summarize my understanding here:
> > > >
> > > > In the GridLU-Arrangements task, there are many possible goal states for any given instruction, each of them should be associated with ground-truth reward 1 (that is used for success rate computation), and other states should give ground-truth reward 0. However, there may be a lot of ambiguity and uncertainty in the natural language instructions, and thus variabilities in the goal states. So there is no (ground-truth) function available that can identify if any given state is the goal or not---except for human annotator. Even though one can find human annotators for evaluation, it is not practical to use them for training the agents.
> > > >
> > > > TLDR: There is no (easy-to-be-)known ground-truth reward function.
> > > >
> > > > Am I right?
> > > >
> > > > Remarks: As I said, I think this is a good paper, and I would like a deep discussion with authors to give a fair evaluation. And I totally understand the authors are eager to convince me to re-evaluate the paper. But before we move on, we should first be on the same page about this above point.

---

> > > > > ### Author Response · Authors · 2018-11-09
> > > > > **Further further clarifications :)**
> > > > >
> > > > > We thank you again for continuing to engage in this discussion and for reading our lengthy response. We truly appreciate your patience and understanding.
> > > > >
> > > > > > In the GridLU-Arrangements task, there are many possible goal states for any given instruction, each of them should be associated with ground-truth reward 1 (that is used for success rate computation), and other states should give ground-truth reward 0.
> > > > >
> > > > > This is correct, although this is true of many tasks and environments across the RL literature (although not all).
> > > > >
> > > > > > However, there may be a lot of ambiguity and uncertainty in the natural language instructions, and thus variabilities in the goal states.
> > > > >
> > > > > The motivation for methods like AGILE is to deal with cases where we want to learn policies that act based on ambiguous or under-specified instructions, for which is may be difficult or impossible to write a reward function within the environment. Instructions in GridLU-Relations may be underspecified, but not to the point where we can’t write a reward function (which admittedly required a surprising amount of code despite the relative simplicity of the instruction language); but this is not the important point, the important point is that we show that a framework which does not have access to the reward (even if the reward function is implemented) and can train a policy as well as one which does, and hence that this framework is useful in environments where it doesn’t have access to the reward *because* the reward function isn’t implemented.
> > > > >
> > > > > > Even though one can find human annotators for evaluation, it is not practical to use them for training the agents.
> > > > >
> > > > > Yes, being able to deal with such scenarios is the motivation for the development of this framework. We emphasise again that this one of the key underlying motivations for all IRL approaches, not just ours.
> > > > >
> > > > > > Am I right?
> > > > >
> > > > > We think we are on the same page now!
> > > > >
> > > > > The question we would put to you, if this addressed your concerns, is (perhaps upon re-reading the relevant sections) is there anything we could clarify to avoid this misunderstanding our ambiguity in portraying our method and its motivation? We have a clear picture of why and how our framework is used, and thus are biased against perceiving unhelpful ambiguities in our presentation, but we are ready to accept that they are there if an informed reader such as yourself had trouble with this aspect of the paper. We are confident that, if there’s anything we could tweak to improve the clarity of the paper, we can do so during the rebuttal period with your help as a byproduct of this discussion.

---

> > > > > > ### Comment · AnonReviewer2 · 2018-11-12
> > > > > > **awesome, please modify your presentation, esp abstract and intro**
> > > > > >
> > > > > >
> > > > > > Great to be on the same page.
> > > > > >
> > > > > > The real problem is: The key motivation/challenge of this work is NOT obvious enough in either abstract or introduction.
> > > > > >
> > > > > > Your rebuttal/clarification was very helpful, and I think you could extract useful content in our back-and-forth to make your points more highlighted.
> > > > > >
> > > > > > Some examples in abstract and introduction may help. Textual descriptions may sometimes be misleading and ambiguous (see---ambiguity is a really big issue :-). You wrote:
> > > > > >
> > > > > > *designing language-conditional reward functions which may not be easily or tractably implemented as the complexity of the environment and the language scales.* (in abstract)
> > > > > >
> > > > > > *This interpreter must be able to evaluate the instruction against environment states to determine what reward must be granted to the agent, and in doing so requires full knowledge (on the part of the designer) of the semantics of the instruction language relative to the environment* (in introduction)
> > > > > >
> > > > > > I could also find other examples like these, but none of them was clear enough that: for any given instruction, there may be many (or even infinite) final states that are correct, and deciding on whether a given one is correct may have to get human involved. But an example might be much clearer.
> > > > > >
> > > > > > You had GridLu Arrangements examples in Figure-2 that could illustrate your point of *many goal states for any instruction*, but not explicit enough (and a little lack of textual description).
> > > > > >
> > > > > > Maybe you can try having one example in your abstract and introduction? (Of course, the one in the abstract should be quite short.) Maybe you can put the most challenging example in your dataset to the introduction, in order to show the challenge and motivate your work. I am not suggesting a major surgery to the paper structure---instead, you only need to adjust the positions of examples and make them more connected to the relevant text.
> > > > > >
> > > > > > I still recommend you to discuss connections to reward-shaping work---it can be light as a few sentences, but it is indeed connected---the simplest setting that you care may have an easy one-to-one instruction->goal state mapping so reward-shaping can be done. In other cases, your work is the solution.
> > > > > >
> > > > > > I would like to raise my evaluation score as I believe these are fairly easy to do in a proper way within the rebuttal period.
> > > > > >
> > > > > > I would also encourage you to compare with reward-shaping in the datasets where a reward-function is indeed accessible---it is interesting to know. But since it is not the focus of the paper, this comparison is optional.

---

> > > > > > > ### Author Response · Authors · 2018-11-15
> > > > > > > **revision uploaded**
> > > > > > >
> > > > > > > Thank you for your recommendations. We are happy to make modifications to the paper to ensure both greater clarity and better links to existing research. We hope you will find them appropriate, and are open to further tweaks if you judge they are necessary.
> > > > > > >
> > > > > > > > The key motivation/challenge of this work is NOT obvious enough in either abstract or introduction.
> > > > > > > > Maybe you can try having one example in your abstract and introduction? [...] you only need to adjust the positions of examples and make them more connected to the relevant text.
> > > > > > >
> > > > > > > We make the reason for this paper clearer by adding an example motivating, in our environment, the design of such reward models, in the new Figure 1 which is placed in the introduction.
> > > > > > >
> > > > > > > > I still recommend you to discuss connections to reward-shaping work---it can be light as a few sentences, but it is indeed connected---the simplest setting that you care may have an easy one-to-one instruction->goal state mapping so reward-shaping can be done. In other cases, your work is the solution.
> > > > > > >
> > > > > > > We add, in the discussion, after “Our analysis of the reward model’s classifications gives a sense of how this is possible; the false positive decisions that it makes early in the training help the policy to start learning.”, the following sentences: “The fact that AGILE’s objective attenuates learning issues due to the sparsity of reward states within episodes in a manner similar to reward prediction suggests that the reward model within AGILE learns some form of shaped reward (Ng et al, 1999), and could serve not only in the cases where a reward function need to be learned in the absence of true reward, but also in cases where environment reward is defined but extremely sparse. As these cases are not the focal area of this study, we note this but leave such investigation for future work.”
> > > > > > >
> > > > > > > Please see the updated PDF for changes.

---

> > > > > > > > ### Comment · AnonReviewer2 · 2018-11-16
> > > > > > > > **intro is much better**
> > > > > > > >
> > > > > > > >
> > > > > > > > Thanks. I think the 1st paragraph of intro (as well as other modified parts) is much better now.
> > > > > > > >
> > > > > > > > A brief example in abstract may still help, e.g., *the complexity of the environment and the language scales---e.g. what is the goal state of "build an L-like shape from red blocks" while there might be infinitely many valid positions and orientations of the target shape.*---but this is not crucial. The Figure-1 is obvious enough.
> > > > > > > >
> > > > > > > > I would like to revise my review and score.

---

> > > ### Author Response · Authors · 2018-11-08
> > > **Additional clarification (part 4)**
> > >
> > > With apologies for the length of our response, we hope this further clarifies things and shows the positioning not just of our work, but of IRL-related methods in general (against which the objections you raise here would also be levelled) with regard to the sparse reward problem and reward shaping. These are fundamentally different methods, that address different and orthogonal problems against the goal of teaching agents to act appropriately within the context of a task. If this discussion of the distinction satisfies you, we hope you will consider revising your score. If not, please let us know where our definitions continue to diverge.
> > >
> > > We would like to conclude on the following conciliatory note: while we believe that your suggestions are based on a conflation of the spare reward problem with the case where reward functions are not available, your remarks do seem to highlight that the methods used to model reward in the latter case might be applicable to the former. This is not an unreasonable research direction, and we have had the opportunity to reflect on it, inspired by our discussion. Notwithstanding its merit, this research direction is separate to both the framing and motivation of this paper, and is not explored within the experiments. We could mention this within the further work section of the paper, but it would feel like taking credit for an idea that came out of this discussion, so our preference would be to not do so. Let us know how you feel about this proposed analysis and resolution.

---

> > > > ### Comment · AnonReviewer2 · 2018-11-08
> > > > **It is of course fine to leave it out of the future work---I only mention it for my review completeness.**
> > > >
> > > > As I said in previous replies, the recommendation of future work is not taken into account of the submission evaluation---I only mention it for the completeness of my review. You are of course free to decide what to include or not :-).

---

> ### Author Response · Authors · 2018-11-26
> **Some final comments**
>
> We thank the reviewer for revising their score, and more importantly, for actively participating in the discussion period. As you well know, with the growth of the field, it has been difficult for conferences to maintain a consistent standard of examination throughout the review process whereby reviewers are guaranteed to be thorough and fair. We appreciate your commitment to the clarification of this paper during this discussion, and without intent of flattery, can assure you that you have been thorough. We ask now, with utmost respect, that you be fair.
>
> You have revised your review to indicate, through your score, that the paper is marginally acceptable. Yet the substance of your review indicated that, thanks in part to the clarifications you assisted us in producing, the paper is good. You kindly state, in support, that “the paper deals with a challenging navigation problem”, that “the idea is neat, and its effectiveness is empirically supported”, that “the paper is well-written”, and “the results are convincing”. On the basis of this review, you conclude “Overall, it is a good paper”.
>
> We appreciate how this request may come across, but after the effort you yourself have committed to the improvement of this paper, we only ask that you either consider assigning it a score which reflects the strength you find in it, or alternatively give us some indication of where it still falls short so as to merit a borderline score. We appreciate, as always, your patience and diligence in this matter, and will respect your decision either way.

---

### Meta-Review · Area_Chair1 · 2018-11-04
**Good work and helpful revisions**

**Confidence:** 4
**Recommendation:** Accept (Poster)

**Metareview:**

Pros:
- The paper is well-written and clear and presented with helpful illustrations and videos.
- The  training methodology seems sound (multiple random seeds etc.)
- The results are encouraging.

Cons:
- There was some concern generally about how this work is positioned relative to related work and the completeness of the related work.  However, the authors have made this clearer in their rebuttal.

There was a considerable amount of discussion between the authors and all reviewers to pin down some unclear aspects of the paper. I believe in the end there was good convergence and I thank both the authors and reviewers for their persistence and dilligence in working through this.  The final paper is much better I think and I recommend acceptance.